# Does the COVID-19 pandemic lockdown affect risk attitudes?—Evidence from rural Thailand

**Hao Luo**[ID]*, Charlotte Reich, Oliver Mußhoff

Department of Agricultural Economics and Rural Development, Georg-August-University of Göttingen, Göttingen, Germany

* hao.luo@uni-goettingen.de

## Abstract

Empirical research provides evidence on changes in individuals' risk attitudes after experiencing exogenous shocks. The global outbreak of the COVID-19 pandemic has had various adverse impacts on economies and households. This study utilizes the COVID-19 pandemic and the accompanying lockdown to explore its impact on risk attitudes in rural Thailand using a difference-in-difference (DiD) approach. Overall, we do not find evidence on considerable changes in the willingness to take risks of rural household members after experiencing a lockdown during the pandemic. However, a significant heterogenous effect is found between individuals working inside and outside the agricultural sector. Individuals working outside the agricultural sector have a statistically significant reduction in their willingness to take risks after experiencing a lockdown. Our study provides additional empirical evidence to understand the impact of shocks on rural households' risk attitudes. This sheds light on how policy designs can better help mitigate downward economic trends following exogenous shocks.

## 1. Introduction

Risk attitudes are a key component of individual decision-making and macroeconomic outcomes. A large number of studies show that risk attitudes have remarkable predictive power for individual portfolio selection, insurance demand, agricultural productivity, occupational choices, migration decisions, and addictive behaviors [1–6]. At the macroeconomic level, risk aversion is strongly correlated with per capita GDP, national income, and income inequality [7–9]. In the context of a global pandemic, risk attitudes are shown to considerably affect compliance behavior regarding public health and social measures, including individuals' decisions to vaccinate [10–12].

In standard economics, risk preferences are assumed to be stable across individuals and over time [13]. However, a growing body of empirical studies provides evidence on the temporal variability of risk attitudes and shows that exogenous negative shocks such as natural disasters, social conflicts, or economic recessions can cause a statistically significant change in individuals' risk attitudes [14, 15] and thus influence economic decision-making. The COVID-19 pandemic constitutes one such extreme event, with potentially substantial impacts on risk attitudes. For example, if risk aversion increases after exposure to the pandemic,

**Data Availability Statement:** The data that support the findings of this study are available from the TVSEP (Thailand Vietnam Socio Economic Panel) research project (https://www.tvsep.de/en/tvsep-data-access). Restrictions apply to the availability

of these data, which were used under license for this study. Data are available from https://www.tvsep.de/en/tvsep-data-access with the permission of Thailand Vietnam Socio Economic Panel research project.

**Funding:** The authors received no specific funding for this work.

**Competing interests:** The authors have declared that no competing interests exist.

individuals may be more likely to comply with preventative measures but may also reduce their investments, which can slow economic recovery. Assessing the temporal stability of risk attitudes can shed additional light on post-pandemic recovery plans. Robust recovery strategies are vital as the global outbreak of the COVID-19 pandemic has hit many countries hard and has placed additional health and financial burdens on households worldwide.

Thailand was the first country outside of China to confirm COVID-19 cases, with the first case reported on January 13, 2020 [16]. Following a peak of confirmed cases on March 22, 2020 [17], the Thai government announced a national emergency, including a national lockdown for more than three months, from the end of March 2020 to the end of June 2020 [18, 19]. These government policies were strict, as reflected by a Stringency Index value of approximately 75 [20]. This index ranges from 0 to 100, with 100 denoting the strictest government responses. In comparison, the index value was around 80 in China during the same period [20]. While a broad array of public health intervention strategies was successful in mitigating coronavirus transmission in the first wave [21, 22], the strict lockdown policies had adversely impacted the economy and household livelihoods [23, 24], leading to a sharp decline in annual GDP growth from 2.3% in 2019 to -6.1% in 2020 [25]. Especially, lockdown stringency and duration are shown to be associated with mental health issues and psychological disorder [26, 27].

Recent literature provides contradictory results on the impacts of the COVID-19 pandemic on risk attitudes. While Angrisani et al. [28], Drichoutis and Nayga [29], and Lohmann et al. [30] do not find considerable changes in risk attitudes due to the pandemic; Bu et al. [31], Graeber et al. [32], and Mussio, Sosa Andrés and Kidwai [33] demonstrate that the pandemic has a statistically significant effect on individuals' risk attitudes. However, the direction in which the pandemic increases or decreases risk aversion is inconclusive.

Most empirical studies on the impacts of the COVID-19 pandemic on risk attitudes rely on student samples or samples from developed countries. However, little is known about the impact of the pandemic on risk attitudes of the rural poor in developing countries, a group which is particularly vulnerable to exogenous shocks, as they most often operate with scarce resources and limited financial safety nets. Assessing the effect of shocks on risk attitudes of these population is important, as Gloede, Menkhoff, and Waibel [34] show that shocks perpetuate vulnerability to poverty via their effect on risk attitudes. This paper explores the impacts of the COVID-19 pandemic on risk attitudes of rural households in Thailand, contributing to empirical literature on exogenous shocks and individual risk aversion, particularly within a non-WEIRD (Western, Educated, Industrialized, Rich, and Democratic) population context. Since the majority of empirical literature provides evidence on changes in risk attitudes in responses to exogenous shocks including the COVID-19 pandemic, we hypothesize that individuals would change their risk aversion after experiencing the COVID-19 pandemic lockdown. Especially, given the heterogeneity in risk attitudes, also in the context of global pandemic [35], we expect a heterogeneous pattern of changes in risk attitudes. Individuals who are more affected by lockdown policies may be more likely to change their willingness to take risks. Utilizing a large-scale data set from longitudinal household surveys allows for the comparison of risk-taking attitudes before and after the COVID-19 lockdown. In addition, geographical variations in the lockdown duration at the village level enables us to attribute changes in risk attitudes to different lockdown lengths.

The remainder of this paper is organized as follows: Section 2 provides background to the existing literature body on risk preferences and shocks. The working data is described in section 3, while section 4 presents our empirical strategy of which the results are shown and discussed in section 5. The paper closes in section 6 with concluding remarks.

## 2. Literature review

### 2.1 The assumption of stable preferences and empirical risk-elicitation measures

The conceptual framework of Stigler and Becker [13] in favor of the stability of preferences over time and across individuals has shaped economics for decades. In their seminal paper, Stigler and Becker [13] argue against incorporating shifts in preferences, proposing that preferences should be considered in different state spaces. As long as choices are state-dependent and state spaces are exogenous to households, any differences in consumption over time and across households can be completely explained by changes in state variables such as prices or incomes, without the need to introduce the unstable preference assumption. Dasgupta et al. [36] empirically test the assertion of Stigler and Becker [13] regarding the stability of preferences under the state-dependent framework and show that while risk attitudes are temporally stable, statistically significant heterogeneity exists among individuals. This echoes empirical findings by Dohmen et al. [37], l'Haridon and Vieider [38], and Vieider, Chmura, and Martinsson [8] demonstrating heterogeneity in risk attitudes across individuals. Besides that, the temporal stability of risk attitudes also lacks empirical robustness. Chuang and Schechter [14] review the results of several empirical papers, concluding that risk attitudes are moderately stable over time, with statistically significant correlation coefficients ranging from 0.13 to 0.55 in most studies. However, the low correlations cannot be entirely attributed to measurement errors and may call into question the empirical validity of the stable risk preference assumption [15]. This prompts the consideration that the degree of preference stability is ultimately an empirical matter [15, 39].

In empirical studies, risk attitudes are typically elicited through incentivized experimental measures or non-incentivized survey questions. A good elicitation method should satisfy both internal validity, implying that different measurements provide a coherent description of the same individual's risk attitudes, and behavioral validity, meaning that measured risk attitudes can predict risk-taking behavior in the real world [15]. Experiments, due to their controlled design and incentives, are considered the gold standard for assessing risk attitudes [40]. Examples include tasks like the lottery-choice tasks designed by Holt and Laury [41], and Gneezy and Potters [42]; the financial investment task designed by Eckel and Grossman [43]; or the Balloon Analogue Risk Task (BART) [44]. By assigning different probabilities to each outcome, i.e. higher expected payoffs correspond to more risks, experimental measures can precisely quantify risks under consideration and empirically test theoretical framework [15]. Yet, conducting incentivized experiments can be expensive and challenging with large and diverse samples, especially in developing or low-education areas [37]. Another strand of methodology is qualitative measures, often used in nationally representative surveys. This approach consists of straightforward questions that, for example, ask participants to assess their willingness to take risks on Likert scales, either generally or in specific domains like driving, health, or finances [37], thus placing less cognitive burden on participants. Dohmen et al. [37] demonstrate that while asking about the willingness to take risks in a specific domain provides a more robust measure for explaining risky behavior in that domain, general self-assessment is the best all-around predictor of different types of risky behavior. While this elicitation cannot reflect the preference parameter in the Arrow–Pratt measure [45, 46] and usually lacks direct monetary incentives, its test-retest stability as well as behavioral validity in predicting actual choices in incentivized experiments and real-life risky behaviors is well-documented across various contexts and countries [47–49]. Moreover, compared with experimental measures, the self-assessment measure has demonstrated greater stability over extended time periods [14] and exhibits better predictive power for real-world outcomes [37].

## 2.2 Adverse shocks and risk attitudes

Empirical research on the temporal stability of risk attitudes is growing rapidly. A large body of empirical literature suggests that covariate shocks such as natural disasters, conflicts, or economic recessions may considerably alter risk aversion [50, 51]. For an overview of empirical studies on the effects of exogenous shocks on risk attitudes see Chuang and Schechter [14], Liebenehm, Schumacher, and Strobl [52], and Gassmann et al. [53].

The research on natural disasters shows conflicting results regarding the direction of their effects. Some studies, including Cameron and Shah [39], Cassar, Healy, and von Kessler [54], Hoang and Le [55], and Liebenehm, Schumacher, and Strobl [52], find that individuals exhibit higher risk aversion after exposure to natural catastrophes. Conversely, other research documents a decline in risk aversion in response to such rare events [56, 57]. The decrease in risk aversion appears to be persistent even five years after events like the Great East Japan Earthquake [58]. However, Eckel, El-Gamal, and Wilson [50] show that evacuees of Hurricane Katrina display higher risk tolerance shortly after the disaster but become moderately more risk-averse one year later. In addition, further studies distinguish risk attitudes in loss and gain domains, revealing less risk-seeking behavior in the loss domain, but less risk aversion or no statistically significant effects in the gain domain [59, 60]. Similarly, conflicts, wars, or nuclear catastrophe may decrease risk aversion [51] or increase risk aversion [61–64], even for those who have not been directly exposed to the traumatic events [65]. Another strand of literature exploring the effects of the macroeconomic environment on risk attitudes provides a relatively consistent conclusion that macroeconomic crises increase individuals' risk aversion [66–70] and that such effects are usually long-lasting [71].

The variation of measurement approaches might be responsible for inconclusive results on whether shocks increase or decrease individuals' risk aversion. A substantial literature body illustrates that attitudes towards risk change when different elicitation methods are employed, in part because participants do not consistently adhere to the same decision strategy across methods [72]. The comparability of findings across empirical studies may only hold true when the same measurement techniques are consistently applied within those studies. Apart from that, empirical studies often face challenges in separating risk attitudes (the curvature of the utility function [73]) from risk perceptions (subjective judgments or rankings of the riskiness of risky options [74]) [75–77]. This could complicate the process of determining changes in risk aversion that can be attributed to shifts in attitudes, perceptions, or a combination of both [52]. This potential bias might be accentuated when utilizing cross-sectional data, given that probability distributions are not uniformly maintained for all respondents. Empirical studies using panel data argue that when respondents are presented with the same explicit stakes and probabilities both before and after the shocks, the observed changes in choices can be attributed to risk attitudes, factoring out the confounder of risk perceptions [58, 67]. Nevertheless, Just and Just [75] point out that the empirical difficulties of disentangling preferences from probability perceptions remains incompletely resolved through experimental methods. Enhanced empirical capability in separately identifying risk attitudes and perceptions could potentially provide more profound insights into the underlying causes behind the conflicting empirical results. In addition, Imas [78] proposes that inconsistent conclusions can be reconciled in distinguishing between realized losses, involving the transfer of money or other forms of value, and paper losses that remain unrealized.

One well-documented channel through which adverse shocks may affect individuals' risk attitudes is emotional responses, such as fear, anger, or sadness [79–82]. Loewenstein et al. [83] propose a risk-as-feelings hypothesis and point out that individuals react to risky situations more emotionally than cognitively and that emotional states often drive behavior. Eckel,

El-Gamal, and Wilson [50] empirically test this emotion-based framework and show that emotional variables have strong explanatory power for changes in risk attitudes after experiencing Hurricane Katrina. Similarly, Hanaoka, Shigeoka, and Watanabe [58] find statistically significant correlations between the self-assessment of symptoms of depression, stress, and sleep problems and variations in risk-taking behavior after exposure to the Great East Japan Earthquake. The fear-based explanation is also supported by Callen et al. [61] and Guiso, Sapienza, and Zingales [67] in the context of social conflicts and financial crisis, respectively.

## 2.3 The COVID-19 pandemic and risk attitudes

The outbreak of the COVID-19 pandemic constitutes a natural experiment for researchers to further investigate the stability of risk attitudes. The pandemic can be considered a natural disaster because of the health risks involved as well as a macroeconomic crisis, since the lockdown policy has severely hampered economic growth [32]. Therefore, empirical studies often use geographical variations in virus prevalence or lockdown stringency as proxies for exposure to the pandemic. The available evidence on the impacts of the COVID-19 pandemic on risk aversion is inconclusive, with most research finding statistically significant changes in risk attitudes after exposure to the pandemic. S1 Table in the supporting information provides an overview of empirical studies on the impacts of the COVID-19 pandemic on risk attitudes.

Similar to empirical findings regarding the impact of other adverse shocks on risk attitudes, it proves challenging to generalize patterns concerning the extent and direction in which the COVID-19 pandemic affects individual willingness to take risks. In the study by Zhang and Palma [84], four elicitation methods are employed, revealing differential results when comparing incentivized behavioral measures and self-assessed measures of risk attitudes in response to the pandemic. Similarly, Adema et al. [85] show that individual risk aversion decreases when employing an incentivized lottery choice, while it increases when utilizing a self-assessed measure in the context of the COVID-19 pandemic. These findings may suggest that the contradictory results may stem from employing different measures to elicit risk attitudes. However, inconsistencies persist even when comparing studies only employing incentivized gamble choice tasks: Some demonstrate the increased risk aversion after exposure to the global pandemic [33, 86], others reveal the decreased risk aversion [53, 85, 87], and the remaining do not discern considerable changes in the risk attitudes [28–30]. Non-incentivized survey questions, i.e., self-reported measures, yield relatively consistent results, with the majority illustrating a decrease in the willingness to take risks [31, 32, 85, 86, 88, 89]. Overall, the results remain inconsistent even when comparing studies using similar elicitation techniques. This may suggest that the noise in risk-elicitation measures, coupled with the presence of diverse contextual factors, contributes to the challenge of obtaining a conclusive understanding of the impact of the global pandemic on individual risk attitudes.

Given the difficulties of data collection during the global pandemic, many empirical studies rely on student samples [29, 30, 53, 87, 90] or online experiment tools, such as Amazon Mechanical Turk (MTurk) [86, 88, 91]. Nonetheless, a few studies use nationally representative household survey data to investigate the stability of risk attitudes under the COVID-19 pandemic. Ikeda, Yamamura, and Tsutsui [92] distinguish between gain and loss domains and show that Japanese households become less sensitive to additional losses, thus increasing the degree of risk tolerance in the loss domain due to the pandemic. In opposition to this, studies using German household surveys offer evidence suggesting a notable increase in risk aversion as a consequence of the COVID-19 pandemic [32, 89]. To the best of our knowledge, the study by Castillo and Hernandez [93] is the only one to explore the relationship between the

COVID-19 pandemic and risk attitudes using a dataset from rural areas in a developing country. The authors find a significant increase in the willingness to take risks among agricultural households in Guatemala when compared to the pre-pandemic period. While their study considers commercial smallholder households, our study broadens perspectives by focusing on the rural poor in general, thus providing further insights into the understanding of risk attitudes under the COVID-19 pandemic.

Our study is similar to that of Gassmann et al. [53], who analyze the impact of the pandemic lockdown on changes in risk attitudes using a sample of students in France. By using incentivized questions in the Multiple Price List format to elicit risk attitudes during and just after the lockdown, the authors find that risk aversion decreases during the lockdown but returns to the initial levels four months after the lockdown, demonstrating the importance of timely and precise political interventions to foster a robust economic recovery. Our study employs a similar measurement of the COVID-19 pandemic, namely the lockdown duration, to analyze its impact on individuals' risk attitudes.

## 3. Data and descriptive statistics

The secondary data used in this study originates from the research project "Poverty dynamics and sustainable development: A long-term panel project in Thailand and Vietnam, 2015–2024", financed by the German Research Foundation (DFG). Data can be accessed at https://www.tvsep.de/en/tvsep-data-access.

The survey initially aims to cover a sample of 2,200 rural households in 220 villages of three provinces in northeastern Thailand, namely Buriram, Ubon Ratchathani, and Nakhon Phanom, where agricultural production dominates and per capita income is close to the poverty line [94]. The sample selection follows a three-stage stratified cluster sampling strategy, with provinces as strata and sub-districts as the primary sampling units. In the three sampled provinces, 110 sub-districts were randomly chosen with the proportional allocation of population density. At the next stages, two villages were selected from each sampled sub-district using the systematic random sampling design, and then 10 households were randomly selected from each village. Therefore, this sample is representative of the rural population in northeastern Thailand.

This study uses the data from the eighth wave household survey conducted between June and August 2019, and the COVID-19 special household as well as village head surveys conducted between November and December 2020. The 2019 household survey covers 2,199 households and includes detailed demographic and socioeconomic characteristics of each household and its members, such as age, gender, education level, household size, and risk attitudes. The COVID-19 special household survey covers 2,141 panel households, providing basic household demographic characteristics and complete information about the effects of the COVID-19 crisis on household financial, consumption, occupational, and health situations. In addition, 220 village heads were interviewed in 2020, allowing for specific information about the lockdown duration, implemented measures, e.g., curfews, school closures, or nongatherings, and the impacts of the COVID-19 crisis at the village level. The COVID-19 special surveys were carried out about five months after the first national lockdown, which lasted from the end of March to June, minimizing recall bias among respondents. The data was anonymized to protect the privacy and confidentiality of individuals, and as such, no ethics approval was required. In the study, we have addressed any ethical considerations related to the use of the data and have taken all necessary steps to ensure that the data was used in a responsible and ethical manner. Informed consent was obtained from participants at the time the data was originally collected.

In total, 1,503 individuals were interviewed both in 2019 and in 2020. Of these individuals, 9 respondents reported a different gender between two surveys, and 27 respondents had a negative or greater than a 3-year age difference between the 2020 and 2019 surveys, which is implausible; these 36 respondents are therefore excluded. Moreover, 51 respondents aged 80 years or older in 2020 are also eliminated to ensure the soundness of the data collected [34]. The final data set for the following analysis is a balanced panel of 1,416 identical respondents from 220 villages. Data processing and analyses were conducted using Stata 15.

Table 1 summarizes the individual characteristics and the COVID-19 pandemic situation of the final sample. On average, respondents were 58 years old, and the majority were females (72.95%) and married (75.07%) in 2019. The average number of years of education was less than 6 years, below the national average of around 9 schooling years in 2019 [95]. The majority of respondents (65.32% in 2019 and 66.17% in 2020) were active in the agricultural sector. Agriculture is often characterized by risk and uncertainty; consequently, 62.57% reported little or great fluctuations in their household income in 2019. In terms of the household structure, on average, each household had about five family members, with a relatively low household dependency ratio of 26.42% in 2019 and 27.38% in 2020. Regarding health status, the majority of respondents (95.34%) felt healthy or able to manage their health problems prior to the pandemic. However, nutritional status remained an issue: Almost half of the respondents (43.64%) were under- or overweight.

The key variables in the analysis are the willingness to take risks as the dependent variable and the lockdown duration as the main explanatory variable. In both waves of each survey, respondents' willingness to take risks is measured through the survey item, "Are you generally a person who is fully prepared to take risks or do you try to avoid taking risks?". Respondents were asked to rate themselves on an ordinal scale from 0 (unwilling to take risks) to 10 (fully prepared to take risks). The lower the score an individual gives to this question, the more risk-averse he or she is. However, due to the type of question asked and the qualitative nature of the scale, a score of 5 (middle category) does not represent risk neutrality [34]. In the household survey 2019, risk attitudes are additionally elicited through a hypothetical investment question asking respondents how much of the 100,000 THB they have just won from a lottery would have been invested in a business that has the equal chance of either doubling or halving the invested amount. A Pearson correlation coefficient of 0.257 with a p-value of 0.000 indicates a highly statistically significant correlation between the self-assessment measure and the hypothetical investment question. Furthermore, using the same data source from different survey years, Hardeweg, Menkhoff, and Waibel [96] validate the simple survey-based risk measure with the Holt and Laury [41] task. This further increases our confidence in the use of the general risk question as a measure of risk attitude and risky behavior among the sampled individuals.

Fig 1 illustrates the distribution of the 1,416 respondents' general willingness to take risks, measured on an 11-point scale. In 2019, prior to the COVID-19 crisis, the average willingness to take risks was 5.466, slightly higher than the middle category of 5. The spike in the histogram is at its middle, with more than a quarter of the respondents rating their willingness to take risks at 5. Another notable feature is the high response rate to the two extreme categories, with around 12% rating themselves at the score of 0 and more than 15% rating themselves at 10. These features are in line with results of other studies using the same panel data source but different years in rural Thailand [34, 96]. Compared to 2019, the responses of 2020 are particularly concentrated on the right side of the figure, suggesting that individuals became more risk-seeking from 2019 to 2020. In 2020, approximately 90% of respondents assessed their willingness to take risks at 5 or above, with an average willingness to take risks of 7.013, much higher than in 2019. This difference in risk attitudes between the two survey years is

**Table 1. Summary statistics of individual characteristics and the COVID-19 pandemic situation.**

| | Observations | 2019 | 2020 | Difference |
|---|---|---|---|---|
| Panel A. Individual level | | | | |
| Willingnes to take risk | 1,416 | 5.466 | 7.013 | 1.547*** |
| | | (3.173) | (2.454) | |
| Age (years) | 1,416 | 58.010 | 59.135 | 1.125*** |
| | | (10.959) | (10.966) | |
| Female[†] | 1,416 | 0.730 | 0.730 | - |
| Married[†] | 1,416 | 0.751 | n.a. | - |
| Education (years) | 1,384 | 5.654 | n.a. | - |
| | | (3.095) | | |
| Agricultural occupation[†] | 1,416 | 0.653 | 0.662 | 0.008 |
| Income fluctuation[†] | 1,412 | 0.626 | n.a. | - |
| Household size | 1,416 | 4.766 | 4.724 | -0.042 |
| | | (1.979) | (2.050) | |
| Dependency ratio | 1,416 | 0.264 | 0.274 | 0.010 |
| Sick[†] | 1,415 | 0.047 | n.a. | - |
| Under—or overweight[†] | 1,416 | 0.436 | n.a. | - |
| Handling stress | 1,415 | 5.302 | 4.035 | -1.266*** |
| | | (1.495) | (1.924) | |
| Getting nervous | 1,415 | 3.475 | 5.708 | 2.233*** |
| | | (1.830) | (1.240) | |
| Compliance with COVID regulations[†] | 1,416 | n.a. | 0.997 | - |
| COVID symptoms[†] | 1,416 | n.a. | 0.062 | - |
| Financial impact of COVID[†] | 1,416 | n.a. | 0.563 | - |
| Panel B. Village level | | | | |
| Declaration of lockdown[†] | 220 | n.a. | 0.086 | - |
| Lockdown duration (days) | 19 | n.a. | 87.421 | - |
| | | | (38.314) | |
| Curfew[†] | 19 | n.a. | 0.842 | - |
| No drinking parties[†] | 19 | n.a. | 0.895 | - |
| Closing schools[†] | 19 | n.a. | 0.632 | - |
| Restricting visiting temples[†] | 19 | n.a. | 0.263 | - |

Notes:

[†] indicates dummy variables, taking the value of 1 if the answer to the corresponding question item is "yes", and 0 otherwise. Means are compared using Wilcoxon-Mann-Whitney's test.

*, **, and *** indicate statistical significance at the 0.10, 0.05, and 0.01 level, respectively.

See S2 Table in the supporting information for the definition of each variable.

Source: Household survey in 2019, COVID-19 special household and village head surveys in 2020, own calculations.

statistically significant, with a z-value of -13.134 and a p-value of 0.000 based on a two-sample Wilcoxon rank-sum (Mann-Whitney) test (see also Table 1).

In an attempt to restrict the possible spread of coronavirus transmission, the Thai government announced an emergency decree and a broad array of public health and social interventions in mid-March of 2020 [20, 21]. As a quick response, all village heads immediately arranged village committee meetings and/or informed their villagers through loudspeakers. Of the 220 villages, 19 villages declared a lockdown. On average, the lockdown lasted approximately 87 days. The shortest lockdown was 41 days, from the end of April to the end of May

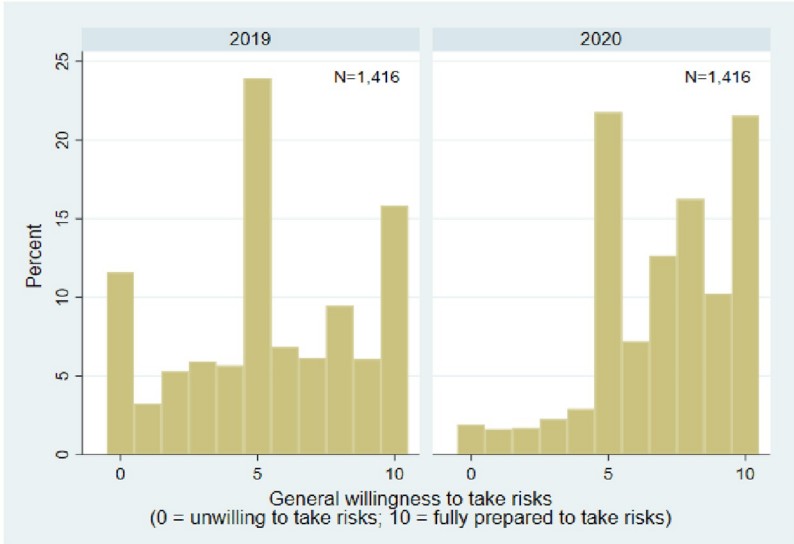

**Fig 1. Histogram of responses to the survey question about general willingness to take risks.** Source: Household survey 2019 and COVID-19 special household survey 2020, own calculations.

2020, and the longest lockdown duration was 193 days, from the end of March to the end of September 2020. During the lockdown, a curfew was in place in 16 villages, drinking parties were forbidden in 17 villages, 12 villages closed schools, and residents of 5 villages were restricted from visiting temples. Compliance was very high, with only 4 out of 1,416 respondents not complying with the COVID-19 regulations during the crisis. As a result, these policies were successful in mitigating the spread of COVID-19 during the first wave of infections in these regions. In fact, only 6.21% of all respondents analyzed in this study showed some symptoms while none tested positive for COVID-19. Nonetheless, around 56% of respondents reported a negative or very negative impact of the COVID-19 crisis on their household's financial situation. Furthermore, mental health, as indicated by how well an individual can handle (the COVID-19 related) stress and how easily he or she gets nervous, showed a statistically significant change between two survey years. After exposure to the first national lockdown, respondents became less able to address stressful situations and they became nervous more easily, suggesting a negative impact of the COVID-19 pandemic on individual mental health. These results are in line with the findings of Sapbamrer et al. [97] who note a decline in farmers' mental health following the lockdown in Thailand. Similarly, Muro, Feliu-Soler, and Castellà [27] report an adverse impact of the lockdown duration on women's wellbeing in Spain, underscoring the universal negative impact of lockdown on individuals' mental health.

## 4. Empirical strategy

This study aims to explore the impacts of the COVID-19 lockdown on risk attitudes by using survey data before and after the first national lockdown in Thailand. The empirical strategy relies on variations in the lockdown duration at the village level while controlling for the unobserved individual time-invariant characteristics using the individual fixed-effects model. The impact of the lockdown is estimated by comparing individuals living in villages without a declared lockdown (control group) and in villages with a declared lockdown (treatment group), assuming that individuals would have the same level of risk aversion in the absence of

the COVID-19 lockdown. Moreover, taking advantage of the panel structure of the data, the impacts can be further isolated by comparing risk attitudes of the same individuals before and after the COVID-19 crisis.

Formally, the following regression equation is estimated:

$$Y_{ikt} = \alpha_{ik} + \beta Duration_{kt} + \delta E_{ikt} + \tau_t + \varepsilon_{ikt}, \tag{1}$$

where $\alpha_{ik}$ represents an individual fixed-effects model netting out the influence of time-invariant characteristics. $Y_{ikt}$ is the survey-based measure of risk attitudes for individual i in village k at time t, and $Duration_{kt}$ is the lockdown duration in days in village k and takes the value of zero before the lockdown. To facilitate the interpretation of outcomes, both variables are normalized to have a mean of zero and a standard deviation of one, as performed by Castillo and Hernandez [93], and Andersen et al. [98]. $E_{ikt}$ denotes individual emotions, i.e. stress and nervousness. The effect of year is also captured, with $\tau_t$, representing the general time trend common to all individuals. $\varepsilon_{ikt}$ represents the error term and is clustered at the village level to account for potential intra-group correlations. The coefficient $\beta$ represents the causal effect of the COVID-19 lockdown on individuals' risk attitudes, depending on the length of the lockdown.

A potential confounding variable in this study is selective migration. For example, after the first lockdown, some households may have moved to villages where the lockdown was shorter or where lockdown measures were not strictly or completely enforced. However, at the time of the pandemic, mobility was limited due to the declared travel restrictions. Moreover, the mobility rate is usually low in traditional agrarian societies. In this sample, only one household changed residence from one village to another between the two surveys. Thus, migration bias is not a concern in this study.

Taking advantage of the panel structure of the data, i.e. repeated observations of the same individual, the effects of the COVID-19 lockdown on risk attitudes can be isolated by treating the individual fixed effect, $\alpha_{ik}$, as a parameter to be estimated [99]. With two periods, Eq (1) is algebraically equivalent to the differencing, formally:

$$\Delta Y_{ikt} = \beta Duration_k + \delta \Delta E_{ikt} + \Delta \tau + \Delta \varepsilon_{ikt}, \tag{2}$$

where the $\Delta$ prefix indicates the change in variables after and before the COVID-19 lockdown. The differencing removes the unobserved individual effects. For notational convenience, we denote $\Delta Duration_{kt}$ as $Duration_k$, because $Duration_{kt}$ takes the value of zero for all observations before the COVID-19 lockdown. This specification is difference-in-difference (DiD), with $Duration_k$ being a continuous variable ($Duration_k$ takes the value of zero if the lockdown in village k was not declared).

The key underlying assumption of the DiD approach is that, in the absence of the treatment, the outcome of treatment and control groups would follow the same trend over time [100]. Although this common-trends assumption cannot be tested directly, comparing the evolution of the average level of risk attitudes between villages with and without a declared lockdown provides some supportive evidence. We use the fifth to eighth wave surveys, carried out in 2013, 2016, 2017, and 2019 respectively, as well as the COVID-19 special survey to perform the pre-trend analysis. Fig 2 plots the average willingness to take risks in villages with and without a declared lockdown. Generally, the willingness to take risks shows a similar pattern over time between villages that declared a lockdown and those that did not, with individuals in villages that declared a lockdown being, on average, slightly more risk-averse. Based on Wilcoxon rank-sum tests, the differences in the point estimates between the two groups are not statistically significant before the COVID-19 crisis, while in 2020 the difference is statistically

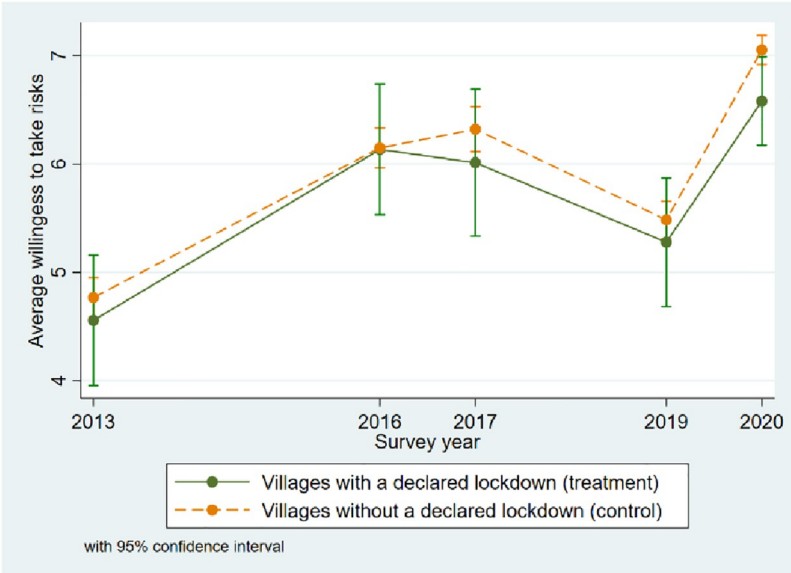

**Fig 2. Average willingness to take risks over time for villages with and without a declared lockdown.** Source: Household surveys in 2013, 2016, 2017, 2019, and COVID-19 special household survey in 2020, own calculations.

significant at the 5% level. In all, there are no systematic differences in the average risk attitudes concerning the duration of the lockdown, supporting the common trends assumption. In addition, the validity of this assumption is further tested by conducting a placebo test in Section 5.3.

## 5. Results and discussions

### 5.1 Main results

Table 2 summarizes the estimates using Eq (2). All specifications include individual fixed effects, netting out the influence of unobserved time-invariant variables. Note that a negative coefficient indicates that risk aversion increases with the duration of the lockdown.

Column (1) displays results for a specification including only individual fixed effects. The willingness to take risks is positively correlated with the lockdown duration and this is statistically significant at the 10% level.

Column (2) presents results, controlling for both individual and year fixed effects. In comparison to the estimate in Column (1), the effect becomes statistically insignificant and changes from positive to negative. The 2020-year fixed effect, which captures general fluctuations in the macroeconomic environment, is large enough to compensate for the effects of the lockdown duration estimated in the model including only the individual fixed effect (Column (1)). The coefficient on the year dummy is positive and highly statistically significant, indicating that individuals are substantially more willing to take risks compared to 2019. Similar large year effects are also found by Graeber et al. [32] using the same self-assessment measure among the German population. They argue that this is unsurprising, given that the COVID-19 pandemic is first and foremost an aggregate shock affecting all individuals.

Since the literature suggests that emotional responses are a potential mechanism through which exogenous shocks affect risk attitudes [82], two individual emotional variables, i.e., handling stress and feeling nervous, are added as controls in Column (3). Both variables are

**Table 2. The impacts of the COVID-19 lockdown on general willingness to take risks.**

| | Dependent Variable: general willingness to take risks | | |
|---|---|---|---|
| | **(1)** | **(2)** | **(3)** |
| Lockdown duration | 0.067* | -0.032 | -0.034 |
| | (0.040) | (0.033) | (0.033) |
| 2020 year effect | | 0.539*** | 0.492*** |
| | | (0.042) | (0.054) |
| Handling stress | | | 0.051* |
| | | | (0.028) |
| Getting nervous | | | 0.071** |
| | | | (0.029) |
| Constant | 0.000*** | -0.269*** | -0.246*** |
| | (0.000) | (0.021) | (0.027) |
| Individual fixed effects | Yes | Yes | Yes |
| Year fixed effects | No | Yes | Yes |
| Controls for emotions | No | No | Yes |
| R-squared within | 0.005 | 0.140 | 0.145 |
| Observations | 2,832 | 2,832 | 2,831 |
| Individuals | 1,416 | 1,416 | 1,416 |

Notes: The table reports the regression estimates of Eq (2) where the dependent variable is the willingness to take risks and the main explanatory variable is the lockdown duration in days at the village level, both normalized to have a mean of zero and a standard deviation of one. Ordinal variables, *Handling stress* and *Getting nervous* are also standardized. Standard errors are in parentheses and clustered at the village level.

*, **, and *** indicate statistical significance at the 0.10, 0.05, and 0.01 level, respectively. The analyses were conducted using the "xtreg" command in Stata software.

Source: Household survey in 2019, COVID-19 special household and village head surveys in 2020, own calculations.

statistically significant in influencing changes in risk attitudes. Individuals who can handle stress well are also more likely to exhibit risk-loving behavior. This is in line with the psychology literature showing that stress can have a statistically significant positive impact on risk aversion [101, 102]. Similar results are demonstrated by Tsutsui and Tsutsui-Kimura [103] in which stress that arose from the COVID-19 pandemic is an influencing factor for changes in risk attitudes. Regarding nervousness, individuals who get nervous easily are more likely to reduce their level of risk aversion. This observation is substantiated by a lab-in-the-field experiment carried out among micro-entrepreneurs in Vietnam. The study reveals that financial worries increase self-reported levels of feeling nervous, leading to reduction in individuals' levels of risk aversion [80].

After adding all controls (see Column (3)), the coefficient on lockdown duration is slightly smaller than zero and statistically insignificant, indicating that there are no considerable changes in the willingness to take risks in response to the lockdown duration. Employing the same self-assessment measures, studies conducted in developed countries have found only modest effect sizes. Graeber et al. [32] find that a one standard deviation increase in the state-level exposure to COVID-19 decreases the willingness to take risks by about 0.057 of a standard deviation in Germany. Also in Germany, Frondel, Osberghaus, and Sommer [89] show that a one standard deviation increase in the self-assessed financial income losses due to the pandemic is associated with a 0.043 standard deviation decrease in the willingness to take risks, and the corresponding impact size is 0.121 standard deviations when considering severe rather than any financial losses. Our results align with a study conducted among agricultural smallholders in rural areas of Guatemala that employed the same risk-elicitation measure [93].

In general, they observe a significant increased willingness to take risks compared to the pre-pandemic period. However, direct exposure to the virus or experiencing community-level confinement attenuate the effects, resulting in increased risk aversion.

## 5.2 Effect heterogeneity

Numerous empirical studies demonstrate the substantial heterogeneity in the response of the willingness to take risks to the COVID-19 pandemic [32, 85]. To account for possible effects of heterogeneity, Eq (2) was rerun including all controls and an additional term that interacts the standardized lockdown duration with a binary variable indicating subgroups under consideration. The following analysis follows the work of Graeber et al. [32] and focuses on pre-existing demographics, socioeconomic and health characteristics that, as demonstrated in the literature, are related to differences in risk attitudes among individuals. The results are presented in Table 3.

Individuals of different genders, ages, marital and family status often differ in their changes in risk attitudes following negative shocks. For example, Hanaoka, Shigeoka, and Watanabe [58] find that men who experienced the 2011 Great East Japan Earthquake become more risk tolerant, while the opposite pattern is observed for women. Graeber et al. [32] show that men and married individuals increase their willingness to take risks after exposure to the COVID-19

**Table 3. Heterogenous effects of the COVID-19 lockdown on general willingness to take risks.**

|  | (1) | (2) | (3) | (4) | (5) | (6) | (7) | (8) | (9) |
|---|---|---|---|---|---|---|---|---|---|
|  | Female | Age $\geq$ 60 | Married | Children < 16 | Secondary education | Agricultural occupation | Income fluctuation | Sick | Under- or overweight |
| Lockdown duration | -0.033 | -0.050 | -0.059 | -0.066 | -0.043 | -0.105*** | -0.071 | -0.029 | -0.011 |
|  | (0.064) | (0.048) | (0.043) | (0.043) | (0.035) | (0.032) | (0.051) | (0.036) | (0.047) |
| 2020 year effect | 0.492*** | 0.493*** | 0.493*** | 0.491*** | 0.502*** | 0.490*** | 0.493*** | 0.495*** | 0.492*** |
|  | (0.054) | (0.054) | (0.054) | (0.054) | (0.054) | (0.054) | (0.054) | (0.054) | (0.054) |
| Handling stress | 0.051* | 0.050* | 0.051* | 0.050* | 0.057** | 0.053* | 0.050* | 0.051* | 0.051* |
|  | (0.027) | (0.027) | (0.027) | (0.027) | (0.028) | (0.027) | (0.028) | (0.028) | (0.027) |
| Getting nervous | 0.071** | 0.070** | 0.071** | 0.072** | 0.066** | 0.072** | 0.073** | 0.070** | 0.071** |
|  | (0.029) | (0.029) | (0.029) | (0.029) | (0.030) | (0.029) | (0.029) | (0.029) | (0.029) |
| Interaction | -0.001 | 0.038 | 0.039 | 0.053 | 0.021 | 0.119* | 0.052 | -0.148 | -0.049 |
|  | (0.054) | (0.060) | (0.055) | (0.049) | (0.094) | (0.062) | (0.058) | (0.187) | (0.051) |
| Constant | -0.246*** | -0.246*** | -0.245*** | -0.245*** | -0.250*** | -0.245*** | -0.248*** | -0.248*** | -0.246*** |
|  | (0.027) | (0.027) | (0.027) | (0.027) | (0.027) | (0.027) | (0.027) | (0.027) | (0.027) |
| Individual fixed effects | Yes | Yes | Yes | Yes | Yes | Yes | Yes | Yes | Yes |
| Year fixed effects | Yes | Yes | Yes | Yes | Yes | Yes | Yes | Yes | Yes |
| Controls for emotions | Yes | Yes | Yes | Yes | Yes | Yes | Yes | Yes | Yes |
| R-squared within | 0.145 | 0.145 | 0.145 | 0.146 | 0.143 | 0.149 | 0.147 | 0.147 | 0.146 |
| Observations | 2831 | 2831 | 2831 | 2831 | 2767 | 2831 | 2823 | 2829 | 2831 |
| Individuals | 1,416 | 1,416 | 1,416 | 1,416 | 1,384 | 1,416 | 1,412 | 1,415 | 1,416 |

Notes: The table reports the results for a regression of the standardized willingness to take risks on the standardized lockdown duration in days at the village level and the respective interaction terms. *Interaction* represents the interaction term between the standardized lockdown duration and a dummy variable indicating subgroups considered. Standard errors are in parentheses and clustered at the village level.

*, **, and *** indicate statistical significance at the 0.10, 0.05, and 0.01 level, respectively.

Source: Household survey 2019, COVID-19 special household and village head surveys 2020, own calculations.

pandemic in a statistically significantly greater degree compared to female and unmarried individuals. Columns (1) to (4) in Table 3 display the corresponding estimates. None of the coefficients on the interaction term is statistically significant, indicating that individuals with these different demographic characteristics do not exhibit differential responses in terms of risk attitudes in our sample. Notably, the marginal change in risk attitudes in response to the lockdown among women and men displays nearly equivalent magnitudes. This result may initially appear unexpected, given the extensive body of empirical research indicating gender risk differences [104, 105] as well as gender differentials in times of the COVID-19 pandemic [106]. While the use of different risk-elicitation methods may influence the manifestation of gender differences in risk taking [107, 108], substantial gender differentials in risk attitudes exist in the pandemic context when employing the self-reported measure [32]. However, a cross-country analysis reports evidence associating the strong gender differentiation in preferences with advanced levels of economic development and gender equality [109]. The study also shows that Thailand exhibits relatively minor gender differences in preferences. Considering the proximity of the surveyed individuals' income to the poverty line, the absence of gender differences in the willingness to take risks might not be as surprising as it appears at first glance.

Similarly, less economically deprived individuals are shown to be more sensitive to the pandemic and to change their risk attitudes more strongly [32]. To account for the effect of heterogeneity in terms of socioeconomic backgrounds, Columns (5) to (7) include individual educational achievement, agricultural occupation, and income fluctuation, respectively. There is a statistically significant difference in the effect of the lockdown duration on risk aversion between individuals in and outside the agricultural sector. On average, as the lockdown duration increases by one standard deviation, the willingness to take risks decreases by 0.105 of a standard deviation for individuals working outside the agricultural sector. Individuals in the non-agricultural sector are more sensitive to the lockdown duration and become statistically significantly more risk-averse, while the lockdown has no statistically significant impact on the willingness to take risks of individuals in the agricultural sector. The result is plausible given that adverse impacts of the lockdown measures, such as social distancing or closure of the workplace, are more pronounced in individuals in the non-agricultural sector [110]. In addition, agriculture is often characterized by risk and uncertainty, especially in rural areas of developing countries. Compared to non-agricultural workers, individuals working in the agricultural sector are already exposed to multiple risks and may have learnt to adopt and navigate unpredictable and hazardous circumstances. As a result, they could be less sensitive to pandemic in terms of changing attitudes towards risk. Other socioeconomic characteristics do not account for a statistically significant heterogeneity of the main result.

Medical literature demonstrates that fatal COVID-19 outcomes are often observed in certain subgroups with pre-existing health problems [111]. Therefore, individuals with certain pre-determined health conditions may exhibit differential responses in terms of risk attitudes, i.e., the high-risk groups of COVID-19 may become more risk averse than the rest of the population. Column (8) and (9) display results considering individuals of different health statuses. After experiencing the lockdown, sick individuals exhibit a much stronger increase in risk aversion than healthy individuals or individuals who can manage their health problems. However, this difference is not statistically significant. In addition, the adjustment in risk attitudes of under- or overweight respondents is not considerably different from others.

## 5.3 Robustness checks

In this section, we perform a series of robustness analyses. Thus far, we have shown that although there is no statistically significant change in risk attitudes among the full sample, the

risk aversion of respondents outside the agricultural sector shows a statistically significant increase after experiencing the first lockdown in rural Thailand. Therefore, the following robustness checks focus on individuals outside the agricultural sector, of which results are presented in Table 4. Column (1) repeats the results of Column (6) in Table 3, taking into account the effect heterogeneity between individuals inside and outside the agricultural sector.

First, the estimated causal inference of the lockdown on risk attitudes relies on the DiD approach. Fig 2 shows that the average willingness to take risks in villages with and without a declared lockdown exhibits approximately parallel, period-specific changes before the pandemic, providing graphical evidence for the credibility of the common trend assumption. To further assess the internal validity of this assumption, we perform a placebo regression based

**Table 4. Robustness checks.**

| | (1) | (2) | (3) | (4) | (5) | (6) |
| --- | --- | --- | --- | --- | --- | --- |
| | Baseline | Placebo | Non-linearity | Outliers | Alternative dependent variable | Alternative independent variable |
| Lockdown duration | -0.105*** | 0.014 | -0.220* | -0.106*** | -0.047** | |
| | (0.032) | (0.037) | (0.122) | (0.032) | (0.019) | |
| Lockdown duration$^2$ | | | 0.017 | | | |
| | | | (0.014) | | | |
| No. of lockdown measures | | | | | | -0.220*** |
| | | | | | | (0.064) |
| Year effect | 0.490*** | -0.159*** | 0.486*** | 0.494*** | 0.207*** | 0.494*** |
| | (0.054) | (0.043) | (0.054) | (0.054) | (0.026) | (0.054) |
| Handling stress | 0.053* | 0.038 | 0.052* | 0.053* | 0.024* | 0.053* |
| | (0.027) | (0.033) | (0.027) | (0.028) | (0.013) | (0.027) |
| Getting nervous | 0.072** | 0.056 | 0.072** | 0.076** | 0.041*** | 0.072** |
| | (0.029) | (0.036) | (0.029) | (0.029) | (0.015) | (0.029) |
| Lockdown duration/measure * agricultural occupation | 0.119* | -0.062 | 0.400*** | 0.118* | 0.060** | 0.241** |
| | (0.062) | (0.063) | (0.144) | (0.062) | (0.024) | (0.100) |
| Lockdown duration$^2$ * agricultural occupation | | | -0.047*** | | | |
| | | | (0.017) | | | |
| Constant | -0.245*** | 0.079*** | -0.231*** | -0.246*** | 0.458*** | -0.240*** |
| | (0.027) | (0.022) | (0.029) | (0.027) | (0.016) | (0.027) |
| Individual fixed effects | Yes | Yes | Yes | Yes | Yes | Yes |
| Year fixed effects | Yes | Yes | Yes | Yes | Yes | Yes |
| Controls for emotions | Yes | Yes | Yes | Yes | Yes | Yes |
| R-squared within | 0.149 | 0.021 | 0.152 | 0.580 | 0.126 | 0.150 |
| Observations | 2,831 | 1,770 | 2,831 | 2,831 | 2,831 | 2,831 |
| Individuals | 1,416 | 885 | 1,416 | 1,416 | 1,416 | 1,416 |

Notes: The table presents results for robustness checks. Column (1) repeats the results from Table 3 Column (6). Column (2) presents the results for the placebo test, i.e. replicating the main analysis on two periods prior to the COVID-19 pandemic (2017 and 2019). Column (3) includes additionally an interaction term of the standardized lockdown duration to account for nonlinear effects. Column (4) presents the results of using the M-estimation to treat outliers. The M-estimation was conducted using the "robreg" command in Stata software. Column (5) presents the results of using a dummy variable, taking the value of 1 if a subject rates his/her willingness to take risks at the five highest categories, and 0 otherwise, as an alternative dependent variable. The estimate was conducted by using the "xtlogit" in Stata software. Column (6) presents the results using the number of lockdown measures implemented at each village, normalized to have mean zero and standard deviation one, as an alternative explanatory variable. Standard errors are in parentheses and clustered at the village level.

*, **, and *** indicate statistical significance at the 0.10, 0.05, and 0.01 level, respectively.

Source: Household surveys in 2017, 2019, COVID-19 special household and village head surveys in 2020, own calculations.

on data from two periods before the lockdown, namely the seventh household survey wave in 2017 and the eighth wave in 2019. If the common trend assumption indeed holds, one would expect that the statistically significant impacts of the lockdown on risk attitudes would not occur in earlier periods. Column (2) in Table 4 presents the results. The coefficient on the lockdown duration is statistically insignificant and has a different sign compared to the baseline results in column (1), which validates the DiD method.

Second, we allow for a nonlinearity in the main independent variable by introducing a squared term of the standardized lockdown duration in the regression equation. Results in Column (2) show that individuals outside the agricultural sector have a strong statistically significant increase in their risk aversion at the beginning of the lockdown, but after a certain length of the lockdown, they become risk tolerant.

Another concern is that results are sensitive to outliers. To address this issue, we follow the M-estimation for regression proposed by Huber [112], which gives less weight to residuals that are more likely to be outliers. The results using M-estimator (Column (3)) remain mostly unchanged both quantitatively and in terms of their statistical significance, suggesting that the main result is not driven by outliers.

Next, the results may be influenced by the way in which risk attitudes are measured. The dependent variable, the willingness to take risks, is an ordinally scaled variable that allows for monotonic increasing transformations. When the empirical analysis relies on models appropriate for quantitative variables, such as the fixed-effects model, such a transformation can lead to contradictory results [113]. To account for this concern, we follow the idea of Graeber et al. [32] and transform the dependent variable into a dummy that is insensitive to monotonic transformations. The dummy variable takes the value of one if a respondent rates his or her willingness to take risks above 5, and zero otherwise. The results are presented in Column (5) and are robust to this alternative construction of the dependent variable.

As a final robustness check, we assess the results by measuring lockdown severity in different terms. In the main analysis, the lockdown duration measured in days is considered a proxy for the lockdown intensity. However, not only the length of the lockdown, but also how many different measures were implemented, such as curfew, school closures, or forbidding gatherings, contribute to the severity of the lockdown. Thus, we use of the number of measures implemented in each village as an alternative indicator for the lockdown stringency. The estimates are displayed in Column (6), where patterns similar to the baseline results can be observed.

## 6. Concluding remarks

A better understanding of the temporal stability of risk attitudes is of great importance in economics given their role in individual decision-making behavior and macroeconomic outcomes. While classical economics assumes risk preferences to be consistent over time and across individuals [13], new empirical literature suggests that negative shocks such as natural disasters, conflicts, wars, or economic crises can considerably alter individuals' risk aversion [14, 15]. The onset of the COVID-19 pandemic has posed threats to many aspects of individuals' lives, therefore giving rise to the question whether individuals' risk attitudes also change considerably as a result of exposure to the pandemic.

This study contributes to empirical literature on the temporal stability of risk attitudes through investigating whether and to what extent experiencing the first lockdown affects individuals' risk attitudes in rural Thailand. Risk attitudes are elicited by a simple survey item whose internal and behavioral validity is well-documented [37, 47]. A high-quality data set was obtained from longitudinal household surveys, allowing for a clear comparison of specific

individuals' willingness to take risks before and after the lockdown. Together with village-level variations in the lockdown duration, causal inference can be established by using a DiD approach.

This study shows that, on average, rural individuals statistically significantly increase their willingness to take risks in 2020 compared to the pre-pandemic period of 2019. However, this increased risk tolerance is not mainly the result of the COVID-19 pandemic lockdown. Instead, the lockdown policies do not considerably change the level of risk attitudes in our sample. As expected, a statistically significant effect heterogeneity is found between individuals inside and outside the agricultural sector. Individuals working outside the agricultural sector experienced a statistically significant reduction in their willingness to take risks after experiencing the lockdown, while individuals working within the agricultural sector do not substantially change their risk aversion. For individuals outside the agricultural sector, the willingness to take risks decreases by approximately 0.105 of a standard deviation with one standard deviation increase in the lockdown duration at the village level. Consistent with other literature, emotional reactions, i.e., stress and nervousness, are found to influence the adjustment in risk attitudes in a statistically significant way. The result of changes in risk attitudes of non-agricultural sector workers is robust to a placebo test, accounting for the nonlinearity in the effects of lockdown, replacing outliers, and alternative constructions of both dependent and explanatory variables.

The findings of this study elicit important implications for both individual decision-making and broader economic outcomes. As the COVID-19 pandemic influenced lives globally, our research sheds light on the dynamic nature of risk attitudes, showing that exposure to the pandemic, such as lockdown, can influence individuals' willingness to take risks. Particularly noteworthy is the variation in this effect among different sectors of the population. The observed increase in risk aversion of individuals working in the non-agricultural sector after exposure to the first national lockdown may have multifaceted consequences. On the one hand, it could lead to a greater willingness to comply with public health and social measures, but on the other hand, it could reduce their likelihood of investing and starting new businesses. Rural households are generally regarded as risk-averse, and this has the potential to lead to suboptimal decision-making and consequently undermine the overall welfare of impoverished rural populations. Increased risk aversion of non-agricultural workers could become an additional barrier to their escape from poverty and thus stagnate long-term growth. Our findings thus underscore the necessity for nuanced policy responses. Policymakers should recognize that pandemic-induced alterations in risk attitudes may exert enduring impacts, potentially influencing compliance behavior and the trajectory of economic recovery.

Possible future research is necessary to explore whether the effects of the pandemic on risk attitudes are persistent or transitory, since the data used in this study was collected only five months after the first national lockdown in Thailand. Furthermore, due to the lack of specific data, it remains unexamined whether changes in risk attitudes will lead to changes in real-life risk-taking behavior, such as gambling, smoking, or virus prevention. This could offer additional valuable insights into understanding individuals' risk attitudes and related behavior and remains a valuable avenue for future studies.

## Supporting information

**S1 Table. Overview of studies investigating the effects of the COVID-19 pandemic on risk attitudes.**
(PDF)

**S2 Table. Variable description.**
(PDF)

## Acknowledgments

We acknowledge the TVSEP (Thailand Vietnam Socio Economic Panel) research project for providing data. We would like to express our gratitude to two anonymous reviewers and the editors of PLOS ONE for their insightful comments.

## Author Contributions

**Conceptualization:** Hao Luo, Charlotte Reich.

**Data curation:** Hao Luo.

**Formal analysis:** Hao Luo.

**Funding acquisition:** Oliver Mußhoff.

**Investigation:** Hao Luo.

**Methodology:** Hao Luo.

**Project administration:** Hao Luo, Charlotte Reich, Oliver Mußhoff.

**Resources:** Charlotte Reich, Oliver Mußhoff.

**Software:** Hao Luo.

**Supervision:** Charlotte Reich, Oliver Mußhoff.

**Validation:** Hao Luo, Charlotte Reich, Oliver Mußhoff.

**Visualization:** Hao Luo.

**Writing – original draft:** Hao Luo.

**Writing – review & editing:** Hao Luo, Charlotte Reich, Oliver Mußhoff.

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
