## [Decision Letter · Decision Letter 0]

3 Aug 2023

PONE-D-23-18250Does the COVID-19 pandemic lockdown affect risk attitudes? - Evidence from rural ThailandPLOS ONE

Dear Dr. Luo,

Thank you for submitting your manuscript to PLOS ONE. After careful consideration, we feel that it has merit but does not fully meet PLOS ONE’s publication criteria as it currently stands. Therefore, we invite you to submit a revised version of the manuscript that addresses the points raised during the review process.

We look forward to receiving your revised manuscript.

Kind regards,

Tae-Young Pak, Ph.D.

Academic Editor

PLOS ONE

Journal Requirements:

2. Please ensure that you have specified a) Did participants provide their written or verbal informed consent to participate in this study?

Reviewers' comments:

Reviewer's Responses to Questions

**Comments to the Author**

1. Is the manuscript technically sound, and do the data support the conclusions?

Reviewer #1: Yes

Reviewer #2: No

2. Has the statistical analysis been performed appropriately and rigorously? 

Reviewer #1: Yes

Reviewer #2: I Don't Know

3. Have the authors made all data underlying the findings in their manuscript fully available?

Reviewer #1: Yes

Reviewer #2: No

4. Is the manuscript presented in an intelligible fashion and written in standard English?

Reviewer #1: Yes

Reviewer #2: Yes

5. Review Comments to the Author

Reviewer #1: The article presents an interesting --although not very novel-- research question regarding the impact of the Covid 19 pandemic on risk preferences. The authors motivate the paper in the literature related to the stability of preferences. I think it is fine to do this, but this discussion is currently too long and unnecessary. Section 2.1 can be significantly reduced and it will not really affect the paper.

Section 2.2 is very effective in conveying the place of the manuscript in the relevant literature and it is probably more relevant than stability of preferences.

One issue that the authors do not mention currently based on the discussion of stability of preferences is that in order to measure stability of preferences similar instruments to measure risk need to be compared. Table 1 provides a review of Covid 19 paper and their findings. It is clear here that there is a wide variation in the instruments used to measure risk, from survey questions (as used by the authors), to lottery gambles, and other instruments. The use of different tools to measure risk may provide some (unintended impact) in the lack of consensus in the literature about the effects of shocks and more specifically Covid 19 on risk preferences. See for example Zhang et al. (2022) for a discussion on this and the use of multiple measures to capture differences of risk taking behavior in different domains. I would suggest the authors frame their results and compare them based on the instrument they use (self reported measures) with the studies in table 1 and potentially other new studies that need to be added to the literature review.

One of the heterogeneity variables that needs to be carefully discussed is gender. The paper is currently almost silent about this. Risk and gender has been extensively studied in previous literature and the paper can elaborate more on the heterogeneity of risk preferences by gender.

Please be careful about the terminology used in the manuscript. In several places the authors make some statements that are not accurate. For example, in the conclusions section (page 20 lines 475-477), the authors state, "This study shows that, on average, rural individuals slightly increase their risk aversion after experiencing the COVID-19 pandemic lockdown in northeastern Thailand. However, this result is not statistically significant at common significance levels." If the result is NOT statistically significant then there are no differences in risk behavior and you have found evidence of stable risk preferences for the overall sample.

After line 459 in the results section, so what are the implications of the results?

References

Zhang, Peilu, and Marco A. Palma. "Stability of risk preferences during COVID-19: Evidence from four measurements." Frontiers in Psychology 12 (2022): 6619.

Reviewer #2: Reviewer’s comments on “Does the COVID-19 pandemic lockdown affect risk attitudes? - Evidence from rural Thailand”

Abstract of this study: This study aims to examine how Covid-19 has changed the risk attitude of Thai residents. To find this out, the authors used 2019 and 2020 data of the research project “Poverty dynamics and sustainable development: A long-term panel project in Thailand and Vietnam, 2015 – 2024,” to estimate how people’s risk attitude changed by lockdown duration (day) in 19 out of 220 villages in 2020. They found that the coefficient of lockdown duration was negative but insignificant. Next, to allow for the possibility that this effect may vary by attribute, the authors included the intersection of various attributes and lockdown duration, and found that none of these were significant, except for the intersection with the agricultural worker dummy, which was significantly positive at 10%. The authors summarize these results as follows: "Our results provide further insights into the impact of shocks on rural households’ risk attitudes. This sheds light on how policy designs can better help mitigate downward economic trends following exogenous shocks.”

Major comments:

1. We observe risk taking behavior, though risk attitude (a type of preference) cannot be directly observed. Therefore, we must estimate the risk attitude from the observed risk-taking behavior. For example, in the arrow-Platt measure, the curvature of the utility function is estimated. Risk taking behavior is the choice between different risk options, which depends on the magnitude of the risks faced and the risk attitude of the individual. Some studies used measures for risk attitude that do not fully eliminate the change in risk, even though risk changes before and after the event (see papers on the 2008 financial crisis). I imagine that this has resulted in a mixed literature in which people have become risk averse and risk tolerant due to mega events. I imagine that the former is more prevalent because changes in risk cannot always be successfully ruled out. This paper reviews a number of papers, and I hope the authors will try to organize the literature from this perspective.

It is a convenient assumption for economists for making theoretical predictions that risk attitude varies across individuals but is stable within individuals over time. Without it, economic theory would be too complex to make useful predictions. But this argument, of course, does not claim that people’s risk-taking behavior is constant. People’s choices change as the magnitude of risk changes, even if risk attitude is stable.

2. In this paper, self-evaluation of one’s risk attitude or risk-taking behavior is used as an indicator of risk attitude. Whether this indicator represents risk attitude or risk-taking behavior still requires careful examination. In this sense, this study is evaluated as adding another example to the confusing situation that the previous literature has presented.

3. This paper reports “negative results” meaning that the key variable of ‘Lockdown duration’ is not significant at all for most of estimations. Please discuss why the key variable was insignificant. I guess that small number of observations of positive duration, only 19 villages (8.3% of total villages), might be the cause. Another possibility is that situation of infection was not really changed by lockdown. If these suppositions are the case, ‘Lockdown duration’ is not an appropriate variable to investigate the effect of Covid-19.

If the authors cannot offer a meaningful hypothesis that explains insignificant ‘Lockdown duration,’ it might be better to focus on 2020 year dummy, which was highly significant rather than ‘Lockdown duration.’ In this case, descriptive facts that Covid-19 was the largest event in Thailand in 2020 may be necessary.

4. The authors claim that the contribution of this study is to examine “the impact of the pandemic on risk attitudes of the rural poor in developing countries, a group which is particularly vulnerable to exogenous shocks.” If so, they may want to offer their hypothesis concerning the impact, and evaluate their results based on the hypothesis.

Minor comments:

5. ‘Handling stress’ and ‘Getting nervous’ seem to be opposite variables. Is it correct that they show the same sign in Table 2?

6. Please show more information, such as histogram, of ‘Lockdown duration’.

6. PLOS authors have the option to publish the peer review history of their article (what does this mean?). If published, this will include your full peer review and any attached files.

Reviewer #1: No

Reviewer #2: No

---

## [Author Response · Author response to Decision Letter 0]

22 Sep 2023

Reviewer #1 

The article presents an interesting --although not very novel-- research question regarding the impact of the Covid 19 pandemic on risk preferences. The authors motivate the paper in the literature related to the stability of preferences. I think it is fine to do this, but this discussion is currently too long and unnecessary. Section 2.1 can be significantly reduced and it will not really affect the paper.

Author response: Thank you for the valuable suggestion. We have considerably reduced Section 2.1 and we believe this has improved the paper significantly. Combined with the third comment regarding the noise of different measures, we have additionally included a short description of the extensively employed risk-elicitation measures in economic literature to lead to a latter discussion on this issue and have renamed Section 2.1 to “The assumption of stable preferences and empirical risk-elicitation measures”. The modified text reads as follows [pp. 4-5 lines 72-114]:

“The conceptual framework of Stigler and Becker (13) in favor of the stability of preferences over time and across individuals has shaped economics for decades. In their seminal paper, Stigler and Becker (13) argue against incorporating shifts in preferences, proposing that preferences should be considered in different state spaces. As long as choices are state-dependent and state spaces are exogenous to households, any differences in consumption over time and across households can be completely explained by changes in state variables such as prices or incomes, without the need to introduce the unstable preference assumption. Dasgupta et al. (36) empirically test the assertion of Stigler and Becker (13) regarding the stability of preferences under the state-dependent framework and show that while risk attitudes are temporally stable, statistically significant heterogeneity exists among individuals. This echoes empirical findings by Dohmen et al. (37), l'Haridon and Vieider (38), and Vieider, Chmura, and Martinsson (8) demonstrating heterogeneity in risk attitudes across individuals. Besides that, the temporal stability of risk attitudes also lacks empirical robustness. Chuang and Schechter (14) review the results of several empirical papers, concluding that risk attitudes are moderately stable over time, with statistically significant correlation coefficients ranging from 0.13 to 0.55 in most studies. However, the low correlations cannot be entirely attributed to measurement errors and may call into question the empirical validity of the stable risk preference assumption (15). This prompts the consideration that the degree of preference stability is ultimately an empirical matter (15,39). 

In empirical studies, risk attitudes are typically elicited through incentivized experimental measures or non-incentivized survey questions. A good elicitation method should satisfy both internal validity, implying that different measurements provide a coherent description of the same individual's risk attitudes, and behavioral validity, meaning that measured risk attitudes can predict risk-taking behavior in the real world (15). Experiments, due to their controlled design and incentives, are considered the gold standard for assessing risk attitudes (40). Examples include tasks like the lottery-choice tasks designed by Holt and Laury (41), and Gneezy and Potters (42); the financial investment task designed by Eckel and Grossman (43); or the Balloon Analogue Risk Task (BART) (44). By assigning different probabilities to each outcome, i.e. higher expected payoffs correspond to more risks, experimental measures can precisely quantify risks under consideration and empirically test theoretical framework (15). Yet, conducting incentivized experiments can be expensive and challenging with large and diverse samples, especially in developing or low-education areas (37). Another strand of methodology is qualitative measures, often used in nationally representative surveys. This approach consists of straightforward questions that, for example, ask participants to assess their willingness to take risks on Likert scales, either generally or in specific domains like driving, health, or finances (37), thus placing less cognitive burden on participants. Dohmen et al. (37) demonstrate that while asking about the willingness to take risks in a specific domain provides a more robust measure for explaining risky behavior in that domain, general self-assessment is the best all-around predictor of different types of risky behavior. While this elicitation cannot reflect the preference parameter in the Arrow–Pratt measure (45,46) and usually lacks direct monetary incentives, its test-retest stability as well as behavioral validity in predicting actual choices in incentivized experiments and real-life risky behaviors is well-documented across various contexts and countries (47–49). Moreover, compared with experimental measures, the self-assessment measure has demonstrated greater stability over extended time periods (14) and exhibits better predictive power for real-world outcomes (37).”

Section 2.2 is very effective in conveying the place of the manuscript in the relevant literature and it is probably more relevant than stability of preferences.

Author response: Thank you for the comment. Building upon the comment from Reviewer 2, we have added a discussion concerning potential explanations for the inconclusive empirical results regarding the impact of shocks on risk attitudes. The added text reads as follows [pp. 6-7 lines 136-155]:

“The variation of measurement approaches might be responsible for inconclusive results on whether shocks increase or decrease individuals’ risk aversion. A substantial literature body illustrates that attitudes towards risk change when different elicitation methods are employed, in part because participants do not consistently adhere to the same decision strategy across methods (72). The comparability of findings across empirical studies may only hold true when the same measurement techniques are consistently applied within those studies. Apart from that, empirical studies often face challenges in separating risk attitudes (the curvature of the utility function (73)) from risk perceptions (subjective judgments or rankings of the riskiness of risky options (74)) (75–77). This could complicate the process of determining changes in risk aversion that can be attributed to shifts in attitudes, perceptions, or a combination of both (52). This potential bias might be accentuated when utilizing cross-sectional data, given that probability distributions are not uniformly maintained for all respondents. Empirical studies using panel data argue that when respondents are presented with the same explicit stakes and probabilities both before and after the shocks, the observed changes in choices can be attributed to risk attitudes, factoring out the confounder of risk perceptions (58,67). Nevertheless, Just and Just (75) point out that the empirical difficulties of disentangling preferences from probability perceptions remains incompletely resolved through experimental methods. Enhanced empirical capability in separately identifying risk attitudes and perceptions could potentially provide more profound insights into the underlying causes behind the conflicting empirical results. In addition, Imas (78) proposes that inconsistent conclusions can be reconciled in distinguishing between realized losses, involving the transfer of money or other forms of value, and paper losses that remain unrealized.”

One issue that the authors do not mention currently based on the discussion of stability of preferences is that in order to measure stability of preferences similar instruments to measure risk need to be compared. Table 1 provides a review of Covid 19 paper and their findings. It is clear here that there is a wide variation in the instruments used to measure risk, from survey questions (as used by the authors), to lottery gambles, and other instruments. The use of different tools to measure risk may provide some (unintended impact) in the lack of consensus in the literature about the effects of shocks and more specifically Covid 19 on risk preferences. See for example Zhang et al. (2022) for a discussion on this and the use of multiple measures to capture differences of risk taking behavior in different domains. I would suggest the authors frame their results and compare them based on the instrument they use (self reported measures) with the studies in table 1 and potentially other new studies that need to be added to the literature review.

Author response: Thank you for the insightful suggestion. As suggested by the reviewer, we have revised the S1 Table in the supporting information by adding new relevant studies. Accordingly, in Section 2.3 literature review on COVID-19 papers, we have added a paragraph to discuss the potential influence of different risk-elicitation measures on the inconsistent results. The added text reads as follows [pp. 8-9 lines 176-192]:

“Similar to empirical findings regarding the impact of other adverse shocks on risk attitudes, it proves challenging to generalize patterns concerning the extent and direction in which the COVID-19 pandemic affects individual willingness to take risks. In the study by Zhang and Palma (84), four elicitation methods are employed, revealing differential results when comparing incentivized behavioral measures and self-assessed measures of risk attitudes in response to the pandemic. Similarly, Adema et al. (85) show that individual risk aversion decreases when employing an incentivized lottery choice, while it increases when utilizing a self-assessed measure in the context of the COVID-19 pandemic. These findings may suggest that the contradictory results may stem from employing different measures to elicit risk attitudes. However, inconsistencies persist even when comparing studies only employing incentivized gamble choice tasks: Some demonstrate the increased risk aversion after exposure to the global pandemic (33,86), others reveal the decreased risk aversion (53,85,87), and the remaining do not discern considerable changes in the risk attitudes (28–30). Non-incentivized survey questions, i.e., self-reported measures, yield relatively consistent results, with the majority illustrating a decrease in the willingness to take risks (31,32,85,86,88,89). Overall, the results remain inconsistent even when comparing studies using similar elicitation techniques. This may suggest that the noise in risk-elicitation measures, coupled with the presence of diverse contextual factors, contributes to the challenge of obtaining a conclusive understanding of the impact of the global pandemic on individual risk attitudes.”

In addition, when interpreting our main results in Section 5.1, we have compared our results with other COVID-19 papers using the same survey-based measure. The revised text reads as follows [p. 18 lines 413-426]:

“After adding all controls (see Column (3)), the coefficient on lockdown duration is slightly smaller than zero and statistically insignificant, indicating that there are no considerable changes in the willingness to take risks in response to the lockdown duration. Employing the same self-assessment measures, studies conducted in developed countries have found only modest effect sizes. Graeber et al. (32) find that a one standard deviation increase in the state-level exposure to COVID-19 decreases the willingness to take risks by about 0.057 of a standard deviation in Germany. Also in Germany, Frondel, Osberghaus, and Sommer (89) show that a one standard deviation increase in the self-assessed financial income losses due to the pandemic is associated with a 0.043 standard deviation decrease in the willingness to take risks, and the corresponding impact size is 0.121 standard deviations when considering severe rather than any financial losses. Our results align with a study conducted among agricultural smallholders in rural areas of Guatemala that employed the same risk-elicitation measure (93). In general, they observe a significant increased willingness to take risks compared to the pre-pandemic period. However, direct exposure to the virus or experiencing community-level confinement attenuate the effects, resulting in increased risk aversion.”

One of the heterogeneity variables that needs to be carefully discussed is gender. The paper is currently almost silent about this. Risk and gender has been extensively studied in previous literature and the paper can elaborate more on the heterogeneity of risk preferences by gender.

Author response: Thank you for pointing this out. We agree with the reviewer’s assessment that gender differences in risk attitudes, namely that men are, on average, more willing to take risks than women, are widely demonstrated in the literature (1,2). Men and women also react differently to exogenous shocks in terms of adjusting their risk attitudes ((3–5)). In our study, we do not find statistically significant gender differences in changes in risk attitudes. One reason could be that gender differences in preferences are more pronounced in countries with higher levels of economic development and gender equality, as shown in a cross-country study (6). We have pointed out this unexpected result and added the possible explanation. The revised text reads as follows [p. 20 lines 443-454]:

“Notably, the marginal change in risk attitudes in response to the lockdown among women and men displays nearly equivalent magnitudes. This result may initially appear unexpected, given the extensive body of empirical research indicating gender risk differences (104,105) as well as gender differentials in times of the COVID-19 pandemic (106). While the use of different risk-elicitation methods may influence the manifestation of gender differences in risk taking (107,108), substantial gender differentials in risk attitudes exist in the pandemic context when employing the self-reported measure (32). However, a cross-country analysis reports evidence associating the strong gender differentiation in preferences with advanced levels of economic development and gender equality (109). The study also shows that Thailand exhibits relatively minor gender differences in preferences. Considering the proximity of the surveyed individuals’ income to the poverty line, the absence of gender differences in the willingness to take risks might not be as surprising as it appears at first glance.”

Please be careful about the terminology used in the manuscript. In several places the authors make some statements that are not accurate. For example, in the conclusions section (page 20 lines 475-477), the authors state, "This study shows that, on average, rural individuals slightly increase their risk aversion after experiencing the COVID-19 pandemic lockdown in northeastern Thailand. However, this result is not statistically significant at common significance levels." If the result is NOT statistically significant then there are no differences in risk behavior and you have found evidence of stable risk preferences for the overall sample.

Author response: Thank you for the comment. In the original manuscript, we wanted to emphasize that although the effect is not statistically significant at the common significance level, the sign of the coefficient on the lockdown duration is negative and its magnitude is not exactly indistinguishable from zero. We have carefully reformulated our interpretation. Given that the magnitude of the coefficient (-0.034) is small and statistically insignificant (other studies using the same survey-based measure find effect sizes of 0.057 (4) or 0.043 (7), but their results are statistically highly significant), we have adjusted the interpretation of the main results as suggested by the reviewer. Accordingly, throughout the manuscript, we have changed the interpretation of the main results as [p. 27 lines 559-562]: 

“This study shows that, on average, rural individuals statistically significantly increase their willingness to take risks in 2020 compared to the pre-pandemic period of 2019. However, this increased risk tolerance is not mainly the result of the COVID-19 pandemic lockdown. Instead, the lockdown policies do not considerably change the level of risk attitudes in our sample.” 

After line 459 in the results section, so what are the implications of the results?

Author response: Thank you for the comment. We have added the description of the implications of the results on [p. 28 lines 573-587]:

“The findings of this study elicit important implications for both individual decision-making and broader economic outcomes. As the COVID-19 pandemic influenced lives globally, our research sheds light on the dynamic nature of risk attitudes, showing that exposure to the pandemic, such as lockdown, can influence individuals' willingness to take risks. Particularly noteworthy is the variation in this effect among different sectors of the population. The observed increase in risk aversion of individuals working in the non-agricultural sector after exposure to the first national lockdown may have multifaceted consequences. On the one hand, it could lead to a greater willingness to comply with public health and social measures, but on the other hand, it could reduce their likelihood of investing and starting new businesses. Rural households are generally regarded as risk-averse, and this has the potential to lead to suboptimal decision-making and consequently undermine the overall welfare of impoverished rural populations. Increased risk aversion of non-agricultural workers could become an additional barrier to their escape from poverty and thus stagnate long-term growth. Our findings thus underscore the necessity for nuanced policy responses. Policymakers should recognize that pandemic-induced alterations in risk attitudes may exert enduring impacts, potentially influencing compliance behavior and the trajectory of economic recovery.”

Reviewer #2

Reviewer’s comments on “Does the COVID-19 pandemic lockdown affect risk attitudes? - Evidence from rural Thailand”

Abstract of this study: This study aims to examine how Covid-19 has changed the risk attitude of Thai residents. To find this out, the authors used 2019 and 2020 data of the research project “Poverty dynamics and sustainable development: A long-term panel project in Thailand and Vietnam, 2015 – 2024,” to estimate how people’s risk attitude changed by lockdown duration (day) in 19 out of 220 villages in 2020. They found that the coefficient of lockdown duration was negative but insignificant. Next, to allow for the possibility that this effect may vary by attribute, the authors included the intersection of various attributes and lockdown duration, and found that none of these were significant, except for the intersection with the agricultural worker dummy, which was significantly positive at 10%. The authors summarize these results as follows: "Our results provide further insights into the impact of shocks on rural households’ risk attitudes. This sheds light on how policy designs can better help mitigate downward economic trends following exogenous shocks.”

Major comments:

1. We observe risk taking behavior, though risk attitude (a type of preference) cannot be directly observed. Therefore, we must estimate the risk attitude from the observed risk-taking behavior. For example, in the arrow-Platt measure, the curvature of the utility function is estimated. Risk taking behavior is the choice between different risk options, which depends on the magnitude of the risks faced and the risk attitude of the individual. Some studies used measures for risk attitude that do not fully eliminate the change in risk, even though risk changes before and after the event (see papers on the 2008 financial crisis). I imagine that this has resulted in a mixed literature in which people have become risk averse and risk tolerant due to mega events. I imagine that the former is more prevalent because changes in risk cannot always be successfully ruled out. This paper reviews a number of papers, and I hope the authors will try to organize the literature from this perspective. It is a convenient assumption for economists for making theoretical predictions that risk attitude varies across individuals but is stable within individuals over time. Without it, economic theory would be too complex to make useful predictions. But this argument, of course, does not claim that people’s risk-taking behavior is constant. People’s choices change as the magnitude of risk changes, even if risk attitude is stable.

Author response: Thank you for the valuable comment. We agree that the empirical challenges of separating risk attitudes from risk perceptions could result in inconclusive results regarding the impact of shocks on risk attitudes. We have considered this issue by introducing a discussion of this alongside other potential explanations. The added text reads as follows [pp. 6-7 lines 136-155]:

“The variation of measurement approaches might be responsible for inconclusive results on whether shocks increase or decrease individuals’ risk aversion. A substantial literature body illustrates that attitudes towards risk change when different elicitation methods are employed, in part because participants do not consistently adhere to the same decision strategy across methods (72). The comparability of findings across empirical studies may only hold true when the same measurement techniques are consistently applied within those studies. Apart from that, empirical studies often face challenges in separating risk attitudes (the curvature of the utility function (73)) from risk perceptions (subjective judgments or rankings of the riskiness of risky options (74)) (75–77). This could complicate the process of determining changes in risk aversion that can be attributed to shifts in attitudes, perceptions, or a combination of both (52). This potential bias might be accentuated when utilizing cross-sectional data, given that probability distributions are not uniformly maintained for all respondents. Empirical studies using panel data argue that when respondents are presented with the same explicit stakes and probabilities both before and after the shocks, the observed changes in choices can be attributed to risk attitudes, factoring out the confounder of risk perceptions (58,67). Nevertheless, Just and Just (75) point out that the empirical difficulties of disentangling preferences from probability perceptions remains incompletely resolved through experimental methods. Enhanced empirical capability in separately identifying risk attitudes and perceptions could potentially provide more profound insights into the underlying causes behind the conflicting empirical results. In addition, Imas (78) proposes that inconsistent conclusions can be reconciled in distinguishing between realized losses, involving the transfer of money or other forms of value, and paper losses that remain unrealized.”

2. In this paper, self-evaluation of one’s risk attitude or risk-taking behavior is used as an indicator of risk attitude. Whether this indicator represents risk attitude or risk-taking behavior still requires careful examination. In this sense, this study is evaluated as adding another example to the confusing situation that the previous literature has presented.

Author response: Thank you and we appreciate the feedback. The risk-elicitation measure used in this study, self-assessment of the willingness to take risks, is extensively utilized in various nationally representative surveys, such as the German Socioeconomic Panel (SOEP) (8). Its test-retest stability as well as behavioral validity in predicting actual choices in incentivized experiments and real-life risky behaviors is well-documented across various contexts and countries (9–11). However, we agree that this is not a perfect measure and comes with the limitation that it cannot reflect the preference parameter in the Arrow–Pratt measure. We have mentioned this limitation [p. 5 lines 102-114]:

“Another strand of methodology is qualitative measures, often used in nationally representative surveys. This approach consists of straightforward questions that, for example, ask participants to assess their willingness to take risks on Likert scales, either generally or in specific domains like driving, health, or finances (37), thus placing less cognitive burden on participants. Dohmen et al. (37) demonstrate that while asking about the willingness to take risks in a specific domain provides a more robust measure for explaining risky behavior in that domain, general self-assessment is the best all-around predictor of different types of risky behavior. While this elicitation cannot reflect the preference parameter in the Arrow–Pratt measure (45,46) and usually lacks direct monetary incentives, its test-retest stability as well as behavioral validity in predicting actual choices in incentivized experiments and real-life risky behaviors is well-documented across various contexts and countries (47–49). Moreover, compared with experimental measures, the self-assessment measure has demonstrated greater stability over extended time periods (14) and exhibits better predictive power for real-world outcomes (37).”

In our data, the self-assessment measure exhibits a highly significant correlation with the hypothetical investment question using the 2019 survey data. In addition, the study by Hardeweg, Menkhoff, and Waibel (12) assesses the external validity of self-assessment methods using a subsample from our dataset [pp. 12-13 lines 269-284]:

“The key variables in the analysis are the willingness to take risks as the dependent variable and the lockdown duration as the main explanatory variable. In both waves of each survey, respondents’ willingness to take risks is measured through the survey item, “Are you generally a person who is fully prepared to take risks or do you try to avoid taking risks?”. Respondents were asked to rate themselves on an ordinal scale from 0 (unwilling to take risks) to 10 (fully prepared to take risks). The lower the score an individual gives to this question, the more risk-averse he or she is. However, due to the type of question asked and the qualitative nature of the scale, a score of 5 (middle category) does not represent risk neutrality (34). In the household survey 2019, risk attitudes are additionally elicited through a hypothetical investment question asking respondents how much of the 100,000 THB they have just won from a lottery would have been invested in a business that has the equal chance of either doubling or halving the invested amount. A Pearson correlation coefficient of 0.257 with a p-value of 0.000 indicates a highly statistically significant correlation between the self-assessment measure and the hypothetical investment question. Furthermore, using the same data source from different survey years, Hardeweg, Menkhoff, and Waibel (96) validate the simple survey-based risk measure with the Holt and Laury (41) task. This further increases our confidence in the use of the general risk question as a measure of risk attitude and risky behavior among the sampled individuals.”

Throughout the manuscript, we have reframed our contribution to be more about providing additional empirical evidence on the research agenda regarding the impact of shocks on risk attitudes [p. 27 lines 552-554]:

“This study contributes to empirical literature on the temporal stability of risk attitudes through investigating whether and to what extent experiencing the first lockdown affects individuals’ risk attitudes in rural Thailand.”

3. This paper reports “negative results” meaning that the key variable of ‘Lockdown duration’ is not significant at all for most of estimations. Please discuss why the key variable was insignificant. I guess that small number of observations of positive duration, only 19 villages (8.3% of total villages), might be the cause. Another possibility is that situation of infection was not really changed by lockdown. If these suppositions are the case, ‘Lockdown duration’ is not an appropriate variable to investigate the effect of Covid-19. If the authors cannot offer a meaningful hypothesis that explains insignificant ‘Lockdown duration,’ it might be better to focus on 2020 year dummy, which was highly significant rather than ‘Lockdown duration.’ In this case, descriptive facts that Covid-19 was the largest event in Thailand in 2020 may be necessary.

Author response: Thank you for the comment. We applied a quasi-experimental method, difference-in-difference, which is also commonly used in other studies to isolate the impacts of shocks on risk attitudes (4,5,13–15). To ensure this method is valid with our data, we carefully checked the crucial assumption of this method, namely the common-trends assumption [pp. 16-17 lines 359-372]:

“The key underlying assumption of the DiD approach is that, in the absence of the treatment, the outcome of treatment and control groups would follow the same trend over time (100). Although this common-trends assumption cannot be tested directly, comparing the evolution of the average level of risk attitudes between villages with and without a declared lockdown provides some supportive evidence. We use the fifth to eighth wave surveys, carried out in 2013, 2016, 2017, and 2019 respectively, as well as the COVID-19 special survey to perform the pre-trend analysis. Figure 2 plots the average willingness to take risks in villages with and without a declared lockdown. Generally, the willingness to take risks shows a similar pattern over time between villages that declared a lockdown and those that did not, with individuals in villages that declared a lockdown being, on average, slightly more risk-averse. Based on Wilcoxon rank-sum tests, the differences in the point estimates between the two groups are not statistically significant before the COVID-19 crisis, while in 2020 the difference is statistically significant at the 5% level. In all, there are no systematic differences in the average risk attitudes concerning the duration of the lockdown, supporting the common trends assumption. In addition, the validity of this assumption is further tested by conducting a placebo test in Section 5.3.”

There is minimal literature documenting that the small number of observations in the treatment (villages implemented a lockdown policy) could result in insignificant results when applying the DID approach. Nevertheless, we randomly selected a subsample from the control groups (villages did not implement a lockdown) to have a balanced control-treatment group and re-run our analysis. The results are robust to the main results in the manuscript, and the main variable of interests remains statistically insignificant. The results are shown in the following table: 

Table R1 The impacts of the COVID-19 lockdown on general willingness to take risks (using a randomly selected subsample)

 (1) (2) (3)

Lockdown duration 0.151 -0.047 -0.056

 (0.091) (0.087) (0.087)

2020 year effect 0.488*** 0.447***

 (0.111) (0.108)

Handling stress 0.099

 (0.067)

Getting nervous 0.087

 (0.063)

Constant -0.000*** -0.244*** -0.223***

 (0.000) (0.055) (0.054)

Individual fixed effects Yes Yes Yes

Year fixed effects No Yes Yes

Controls for emotions No No Yes

R-squared within 0.027 0.093 0.109

Observations 486 486 486

Individuals 243 243 243

Notes: The table reports the regression estimates of Equation (2) where the dependent variable is the willingness to take risks and the main explanatory variable is the lockdown duration in days at the village level, both normalized to have a mean of zero and a standard deviation of one. Ordinal variables, Handling stress and Getting nervous are also standardized. Standard errors are in parentheses and clustered at the village level. *, **, and *** indicate statistical significance at the 0.10, 0.05, and 0.01 level, respectively. The analyses were conducted using the “xtreg” command in Stata software. 

Source: Household survey in 2019, COVID-19 special household and village head surveys in 2020, own calculations. 

Second, lockdown policies could change the situation of infection on the one hand, but also local economic situations on the other hand, as we argued in the manuscript [p. 8 lines 168-170]:

“The pandemic can be considered a natural disaster because of the health risks involved as well as a macroeconomic crisis, since the lockdown policy has severely hampered economic growth (32).”

There is no number of infection cases available at the village level, only at the county or country level. Nonetheless, lockdown policies have successfully reduced infection in Thailand (16,17). In our sample, lockdown has adversely impacted individuals’ financial and mental health situations [p. 14 lines 299-320]:

“In an attempt to restrict the possible spread of coronavirus transmission, the Thai government announced an emergency decree and a broad array of public health and social interventions in mid-March of 2020 (20,21). As a quick response, all village heads immediately arranged village committee meetings and/or informed their villagers through loudspeakers. Of the 220 villages, 19 villages declared a lockdown. On average, the lockdown lasted approximately 87 days. The shortest lockdown was 41 days, from the end of April to the end of May 2020, and the longest lockdown duration was 193 days, from the end of March to the end of September 2020. During the lockdown, a curfew was in place in 16 villages, drinking parties were forbidden in 17 villages, 12 villages closed schools, and residents of 5 villages were restricted from visiting temples. Compliance was very high, with only 4 out of 1,416 respondents not complying with the COVID-19 regulations during the crisis. As a result, these policies were successful in mitigating the spread of COVID-19 during the first wave of infections in these regions. In fact, only 6.21% of all respondents analyzed in this study showed some symptoms while none tested positive for COVID-19. Nonetheless, around 56% of respondents reported a negative or very negative impact of the COVID-19 crisis on their household’s financial situation. Furthermore, mental health, as indicated by how well an individual can handle (the COVID-19 related) stress and how easily he or she gets nervous, showed a statistically significant change between two survey years. After exposure to the first national lockdown, respondents become less able to address stressful situations and they become nervous more easily, suggesting a negative impact of the COVID-19 pandemic on individual mental health. These results are in line with the findings of Sapbamrer et al. (97) who note a decline in farmers’ mental health following the lockdown in Thailand. Similarly, Muro, Feliu-Soler, and Castellà (27) report an adverse impact of the lockdown duration on women’s wellbeing in Spain, underscoring the universal negative impact of lockdown on individuals’ mental health.”

Third, we believe that lockdown duration is a better indicator for the COVID- 19 than the 2020-year dummy. The global pandemic was not the only significant event in Thailand in 2020. Another remarkable occurrence was the widespread youth-led democracy protest movements against the government and advocating for reform of the monarchy, which began in February 2020 (18). We cannot factor out the influence of other big events and difficult to attribute impacts completely to the COVID-19 pandemic when focusing the 2020-year dummy. Nonetheless, we have pointed out [p. 27 lines 559-562]:

“This study shows that, on average, rural individuals statistically significantly increase their willingness to take risks in 2020 compared to the pre-pandemic period of 2019. However, this increased risk tolerance is not mainly the result of the COVID-19 pandemic lockdown. Instead, the lockdown policies do not considerably change the level of risk attitudes in our sample.”

Last but not least, we have added more possible explanations for the only significant impacts among nonagricultural workers [p. 22 lines 464-477]:

“There is a statistically significant difference in the effect of the lockdown duration on risk aversion between individuals in and outside the agricultural sector. On average, as the lockdown duration increases by one standard deviation, the willingness to take risks decreases by 0.105 of a standard deviation for individuals working outside the agricultural sector. Individuals in the non-agricultural sector are more sensitive to the lockdown duration and become statistically significantly more risk-averse, while the lockdown has no statistically significant impact on the willingness to take risks of individuals in the agricultural sector. The result is plausible given that adverse impacts of the lockdown measures, such as social distancing or closure of the workplace, are more pronounced in individuals in the non-agricultural sector (110). In addition, agriculture is often characterized by risk and uncertainty, especially in rural areas of developing countries. Compared to non-agricultural workers, individuals working in the agricultural sector are already exposed to multiple risks and may have learnt to adopt and navigate unpredictable and hazardous circumstances. As a result, they could be less sensitive to pandemic in terms of changing attitudes towards risk.”

4. The authors claim that the contribution of this study is to examine “the impact of the pandemic on risk attitudes of the rural poor in developing countries, a group which is particularly vulnerable to exogenous shocks.” If so, they may want to offer their hypothesis concerning the impact, and evaluate their results based on the hypothesis.

Author response: Thank you for the suggestion. We have introduced a hypothesis regarding the influence of exogenous shocks on the risk attitudes of the rural poor [p. 3 lines 50-63]:

“…, little is known about the impact of the pandemic on risk attitudes of the rural poor in developing countries, a group which is particularly vulnerable to exogenous shocks, as they most often operate with scarce resources and limited financial safety nets. Assessing the effect of shocks on risk attitudes of these population is important, as Gloede, Menkhoff, and Waibel (34) show that shocks perpetuate vulnerability to poverty via their effect on risk attitudes. This paper explores the impacts of the COVID-19 pandemic on risk attitudes of rural households in Thailand, contributing to empirical literature on exogenous shocks and individual risk aversion, particularly within a non-WEIRD (Western, Educated, Industrialized, Rich, and Democratic) population context. Since the majority of empirical literature provides evidence on changes in risk attitudes in responses to exogenous shocks including the COVID-19 pandemic, we hypothesize that individuals would change their risk aversion after experiencing the COVID-19 pandemic lockdown. Especially, given the heterogeneity in risk attitudes, also in the context of global pandemic (35), we expect a heterogeneous pattern of changes in risk attitudes. Individuals who are more affected by lockdown policies may be more likely to change their willingness to take risks.”

We have carefully assessed our results in relation to this hypothesis, providing insight into the impact of such shocks on risk attitudes [pp. 27-28 lines 559-568]:

“This study shows that, on average, rural individuals statistically significantly increase their willingness to take risks in 2020 compared to the pre-pandemic period of 2019. However, this increased risk tolerance is not mainly the result of the COVID-19 pandemic lockdown. Instead, the lockdown policies do not considerably change the level of risk attitudes in our sample. As expected, a statistically significant effect heterogeneity is found between individuals inside and outside the agricultural sector. Individuals working outside the agricultural sector experienced a statistically significant reduction in their willingness to take risks after experiencing the lockdown, while individuals working within the agricultural sector do not substantially change their risk aversion. For individuals outside the agricultural sector, the willingness to take risks decreases by approximately 0.105 of a standard deviation with one standard deviation increase in the lockdown duration at the village level.” 

We have also compared our results with empirical studies using the same risk-elicitation measure [p. 19 lines 413-426]:

“After adding all controls (see Column (3)), the coefficient on lockdown duration is slightly smaller than zero and statistically insignificant, indicating that there are no considerable changes in the willingness to take risks in response to the lockdown duration. Employing the same self-assessment measures, studies conducted in developed countries have found only modest effect sizes. Graeber et al. (32) find that a one standard deviation increase in the state-level exposure to COVID-19 decreases the willingness to take risks by about 0.057 of a standard deviation in Germany. Also in Germany, Frondel, Osberghaus, and Sommer (89) show that a one standard deviation increase in the self-assessed financial income losses due to the pandemic is associated with a 0.043 standard deviation decrease in the willingness to take risks, and the corresponding impact size is 0.121 standard deviations when considering severe rather than any financial losses. Our results align with a study conducted among agricultural smallholders in rural areas of Guatemala that employed the same risk-elicitation measure (93). In general, they observe a significant increased willingness to take risks compared to the pre-pandemic period. However, direct exposure to the virus or experiencing community-level confinement attenuate the effects, resulting in increased risk aversion.”

Additionally, we have discussed the implications of our findings for policymakers. Developing policies that enhance the economic conditions of the rural poor and integrating our insights into how risk attitudes shape economic behaviors are crucial steps for fostering economic development and resilience in the face of exogenous shocks. The revised text reads as follows [p. 28 lines 573-587]:

“The findings of this study elicit important implications for both individual decision-making and broader economic outcomes. As the COVID-19 pandemic influenced lives globally, our research sheds light on the dynamic nature of risk attitudes, showing that exposure to the pandemic, such as lockdown, can influence individuals' willingness to take risks. Particularly noteworthy is the variation in this effect among different sectors of the population. The observed increase in risk aversion of individuals working in the non-agricultural sector after exposure to the first national lockdown may have multifaceted consequences. On one hand, it could lead to a greater willingness to comply with public health and social measures, but on the other hand, it could reduce their likelihood of investing and starting new businesses. Rural households are generally regarded as risk-averse, and this has the potential to lead to suboptimal decision-making and consequently undermine the overall welfare of impoverished rural populations. Increased risk aversion of non-agricultural workers could become an additional barrier to their escape from poverty and thus stagnate long-term growth. Our findings thus underscore the necessity for nuanced policy responses. Policymakers should recognize that pandemic-induced alterations in risk attitudes may exert enduring impacts, potentially influencing compliance behavior and the trajectory of economic recovery.”

Minor comments:

5. ‘Handling stress’ and ‘Getting nervous’ seem to be opposite variables. Is it correct that they show the same sign in Table 2?

Author response: Thank you for the comment. We agree that the coefficients on “handing stress” and “getting nervous” should be expected to be opposite. We carefully checked the variable construction, Stata codes and results. In our sample, coefficients on these two variables have the same positive sign, namely that both individuals who can handle stress well and those that get nervous easily are more likely to reduce their level of risk aversion. We have added findings from other studies to support our results. The revised text reads as follows [pp. 18-19 lines 401-412]:

“Since the literature suggests that emotional responses are a potential mechanism through which exogenous shocks affect risk attitudes (82), two individual emotional variables, i.e., handling stress and feeling nervous, are added as controls in Column (3). Both variables are statistically significant in influencing changes in risk attitudes. Individuals who can handle stress well are also more likely to exhibit risk-loving behavior. This is in line with the psychology literature showing that stress can have a statistically significant positive impact on risk aversion (101,102). Similar results are demonstrated by Tsutsui and Tsutsui-Kimura (103) in which stress that arose from the COVID-19 pandemic is an influencing factor for changes in risk attitudes. Regarding nervousness, individuals who get nervous easily are more likely to reduce their level of risk aversion. This observation is substantiated by a lab-in-the-field experiment carried out among micro-entrepreneurs in Vietnam. The study reveals that financial worries increase self-reported levels of feeling nervous, leading to reduction in individuals’ levels of risk aversion (80). ”

6. Please show more information, such as histogram, of ‘Lockdown duration’.

Author response: Thank you for the suggestion. We have added more general background information on the lockdown in Thailand [p. 2 lines 32-42]:

“Thailand was the first country outside of China to confirm COVID-19 cases, with the first case reported on January 13, 2020 (16). Following a peak of confirmed cases on March 22, 2020 (17), the Thai government announced a national emergency, including a national lockdown for more than three months, from the end of March 2020 to the end of June 2020 (20,21). These government policies were strict, as reflected by a Stringency Index value of approximately 75 (22). This index ranges from 0 to 100, with 100 denoting the strictest government responses. In comparison, the index value was around 80 in China during the same period (22). While a broad array of public health intervention strategies was successful in mitigating coronavirus transmission in the first wave (18,19), the strict lockdown policies had adversely impacted the economy and household livelihoods (23,24), leading to a sharp decline in annual GDP growth from 2.3% in 2019 to -6.1% in 2020 (25). Especially, lockdown stringency and duration are shown to be associated with mental health issues and psychological disorder (26,27). ”

Regarding the lockdown situation in our sample, details concerning the lockdown duration and implemented measures have been presented in Panel B of Table 1 [p. 12 lines 262-268]:

Table 1 Summary statistics of individual characteristics and the COVID-19 pandemic situation

 Observations 2019 2020 Difference

Panel A. Individual level

… …

Panel B. Village level

〖Declaration of lockdown〗^† 220 n.a. 0.086 -

Lockdown duration (days) 19 n.a. 87.421

(38.314) -

Curfew^† 19 n.a. 0.842 -

〖No drinking parties〗^† 19 n.a. 0.895 -

〖Closing schools〗^† 19 n.a. 0.632 -

〖Restricting visiting temples〗^† 19 n.a. 0.263 -

Notes: † indicates dummy variables. Means are compared using Wilcoxon-Mann-Whitney’s test. *, **, and *** indicate statistical significance at the 0.10, 0.05, and 0.01 level, respectively. See S2 Table in the supporting information for the definition of each variable.

Source: Household survey in 2019, COVID-19 special household and village head surveys in 2020, own calculations. 

We have also described in the text [p. 14 lines 299-320]: 

“In an attempt to restrict the possible spread of coronavirus transmission, the Thai government announced an emergency decree and a broad array of public health and social interventions in mid-March of 2020 (20,21). As a quick response, all village heads immediately arranged village committee meetings and/or informed their villagers through loudspeakers. Of the 220 villages, 19 villages declared a lockdown. On average, the lockdown lasted approximately 87 days. The shortest lockdown was 41 days, from the end of April to the end of May 2020, and the longest lockdown duration was 193 days, from the end of March to the end of September 2020. During the lockdown, a curfew was in place in 16 villages, drinking parties were forbidden in 17 villages, 12 villages closed schools, and residents of 5 villages were restricted from visiting temples. Compliance was very high, with only 4 out of 1,416 respondents not complying with the COVID-19 regulations during the crisis. As a result, these policies were successful in mitigating the spread of COVID-19 during the first wave of infections in these regions. In fact, only 6.21% of all respondents analyzed in this study showed some symptoms while none tested positive for COVID-19. Nonetheless, around 56% of respondents reported a negative or very negative impact of the COVID-19 crisis on their household’s financial situation. Furthermore, mental health, as indicated by how well an individual can handle (the COVID-19 related) stress and how easily he or she gets nervous, showed a statistically significant change between two survey years. After exposure to the first national lockdown, respondents become less able to address stressful situations and they become nervous more easily, suggesting a negative impact of the COVID-19 pandemic on individual mental health. These results are in line with the findings of Sapbamrer et al. (97) who note a decline in farmers’ mental health following the lockdown in Thailand. Similarly, Muro, Feliu-Soler, and Castellà (27) report an adverse impact of the lockdown duration on women’s wellbeing in Spain, underscoring the universal negative impact of lockdown on individuals’ mental health.”

However, due to the data structure, in which over 90% of villages did not implement a lockdown, illustration of lockdown duration for all villages in a histogram may be dominated by the bar with a value of zero (see the histogram below). While we can provide a visual representation only for villages with lockdown implementations, that does not lead to more information than when presented in Table 1 and in text. Therefore, we respectfully do not incorporate this suggestion in the manuscript. 

References

1. Croson R, Gneezy U. Gender Differences in Preferences. J Econ Lit. 2009 Jun;47(2):448–74. 

2. Eckel CC, Grossman PJ. Chapter 113 Men, Women and Risk Aversion: Experimental Evidence. In: Plott CR, Smith VL, editors. Handbook of Experimental Economics Results [Internet]. Elsevier; 2008 [cited 2023 Aug 17]. p. 1061–73. Available from: https://www.sciencedirect.com/science/article/pii/S1574072207001138

3. Belot M, Müller S, Rau HA, Schwieren C. Editorial: Gender Differentials in Times of COVID-19. Front Psychol [Internet]. 2022 [cited 2023 Aug 17];13. Available from: https://www.frontiersin.org/articles/10.3389/fpsyg.2022.901087

4. Graeber D, Schmidt U, Schroeder C, Seebauer J. The effect of a major pandemic on risk preferences - Evidence from exposure to COVID-19 [Internet]. Rochester, NY: Rochester, NY; 2020 Nov [cited 2023 Feb 13]. Report No.: 3724461. Available from: https://papers.ssrn.com/abstract=3724461

5. Hanaoka C, Shigeoka H, Watanabe Y. Do Risk Preferences Change? Evidence from the Great East Japan Earthquake. Am Econ J Appl Econ. 2018 Apr;10(2):298–330. 

6. Falk A, Hermle J. Relationship of gender differences in preferences to economic development and gender equality. Science. 2018 Oct 19;362(6412):eaas9899. 

7. Frondel M, Osberghaus D, Sommer S. Corona and the stability of personal traits and preferences: Evidence from Germany [Internet]. Mannheim, Germany: Leibniz-Zentrum für Europäische Wirtschaftsforschung (ZEW), Mannheim, Germany; 2021 [cited 2023 Feb 13]. Report No.: 21–029. Available from: https://www.econstor.eu/handle/10419/232943

8. Wagner GG, Frick JR, Schupp J. The German Socio-Economic Panel Study (SOEP) - Evolution, Scope and Enhancements [Internet]. Rochester, NY; 2007 [cited 2023 Sep 15]. Available from: https://papers.ssrn.com/abstract=1028709

9. Ding X, Hartog J, Sun Y. Can we measure individual risk attitudes in a survey? [Internet]. Rochester, NY: Rochester, NY; 2010 Mar [cited 2023 Feb 13]. Report No.: 1570425. Available from: https://papers.ssrn.com/abstract=1570425

10. Falk A, Becker A, Dohmen T, Huffman D, Sunde U. The Preference Survey Module: A Validated Instrument for Measuring Risk, Time, and Social Preferences. Manag Sci. 2023 Apr;69(4):1935–50. 

11. Lönnqvist JE, Verkasalo M, Walkowitz G, Wichardt PC. Measuring individual risk attitudes in the lab: Task or ask? An empirical comparison. J Econ Behav Organ. 2015 Nov 1;119:254–66. 

12. Hardeweg B, Menkhoff L, Waibel H. Experimentally Validated Survey Evidence on Individual Risk Attitudes in Rural Thailand. Econ Dev Cult Change. 2013;61(4):859–88. 

13. Cameron L, Shah M. Risk-taking behavior in the wake of natural disasters. J Hum Resour. 2015;50(2):484–515. 

14. Lohmann PM, Gsottbauer E, You J, Kontoleon A. Anti-social behaviour and economic decision-making: Panel experimental evidence in the wake of COVID-19. J Econ Behav Organ. 2023 Feb 1;206:136–71. 

15. Sakha S. Determinants of risk aversion over time: Experimental evidence from rural Thailand. J Behav Exp Econ. 2019 Jun 1;80:184–98. 

16. Dechsupa S, Assawakosri S, Phakham S, Honsawek S. Positive impact of lockdown on COVID-19 outbreak in Thailand. Travel Med Infect Dis. 2020 Jul 1;36:101802. 

17. Wongtanasarasin W, Srisawang T, Yothiya W, Phinyo P. Impact of national lockdown towards emergency department visits and admission rates during the COVID-19 pandemic in Thailand: A hospital-based study. Emerg Med Australas. 2021;33(2):316–23. 

18. Lippert P. Thailand’s Youth between Protest and Repression - Rosa-Luxemburg-Stiftung. 2021 Nov 10 [cited 2023 Aug 25]; Available from: https://www.rosalux.de/en/news/id/45343/thailands-youth-between-protest-and-repression

---

## [Decision Letter · Decision Letter 1]

2 Oct 2023

Does the COVID-19 pandemic lockdown affect risk attitudes? - Evidence from rural Thailand

PONE-D-23-18250R1

Dear Dr. Luo,

We’re pleased to inform you that your manuscript has been judged scientifically suitable for publication and will be formally accepted for publication once it meets all outstanding technical requirements.

Kind regards,

Tae-Young Pak, Ph.D.

Academic Editor

PLOS ONE

Additional Editor Comments (optional):

Reviewers' comments:

Reviewer's Responses to Questions

**Comments to the Author**

1. If the authors have adequately addressed your comments raised in a previous round of review and you feel that this manuscript is now acceptable for publication, you may indicate that here to bypass the “Comments to the Author” section, enter your conflict of interest statement in the “Confidential to Editor” section, and submit your "Accept" recommendation.

Reviewer #1: All comments have been addressed

2. Is the manuscript technically sound, and do the data support the conclusions?

Reviewer #1: Yes

3. Has the statistical analysis been performed appropriately and rigorously? 

Reviewer #1: Yes

4. Have the authors made all data underlying the findings in their manuscript fully available?

Reviewer #1: (No Response)

5. Is the manuscript presented in an intelligible fashion and written in standard English?

Reviewer #1: Yes

6. Review Comments to the Author

Reviewer #1: (No Response)

7. PLOS authors have the option to publish the peer review history of their article (what does this mean?). If published, this will include your full peer review and any attached files.

Reviewer #1: No

---

## [Editor Report · Acceptance letter]

10 Oct 2023

PONE-D-23-18250R1 

Does the COVID-19 pandemic lockdown affect risk attitudes? - Evidence from rural Thailand 

Dear Dr. Luo:

I'm pleased to inform you that your manuscript has been deemed suitable for publication in PLOS ONE. Congratulations! Your manuscript is now with our production department. 

Kind regards, 

on behalf of

Tae-Young Pak 

Academic Editor

PLOS ONE